# CERTIFIED ROBUSTNESS UNDER BOUNDED LEVENSHTEIN DISTANCE

**Elias Abad Rocamora**[EPFL]**, Grigorios G. Chrysos**[W]**, Volkan Cevher**[EPFL]

[EPFL] : LIONS - École Polytechnique Fédérale de Lausanne, Switzerland

[W] : Department of Electrical and Computer Engineering, University of Wisconsin-Madison, USA

{elias.abadrocamora, volkan.cevher}@epfl.ch, chrysos@wisc.edu

## ABSTRACT

Text classifiers suffer from small perturbations, that if chosen adversarially, can dramatically change the output of the model. Verification methods can provide robustness certificates against such adversarial perturbations, by computing a sound lower bound on the robust accuracy. Nevertheless, existing verification methods incur in prohibitive costs and cannot practically handle Levenshtein distance constraints. We propose the first method for computing the Lipschitz constant of convolutional classifiers with respect to the Levenshtein distance. We use these Lipschitz constant estimates for training 1-Lipschitz classifiers. This enables computing the certified radius of a classifier in a single forward pass. Our method, `LipsLev`, is able to obtain $38.80\%$ and $13.93\%$ verified accuracy at distance $1$ and $2$ respectively in the AG-News dataset, while being $4$ orders of magnitude faster than existing approaches. We believe our work can open the door to more efficient verification in the text domain.

## 1 INTRODUCTION

Despite the impressive performance of Natural Language Processing (NLP) models (Sutskever et al., 2014; Zhang et al., 2015; Devlin et al., 2019), simple corruptions like typos or synonym substitutions are able to dramatically change the prediction of the model (Belinkov and Bisk, 2018; Alzantot et al., 2018). With newer attacks in NLP becoming stronger (Hou et al., 2023), verification methods become relevant for providing future-proof robustness certificates (Liu et al., 2021).

Constraints on the Levenshtein distance (Levenshtein et al., 1966) provide a good description of the perturbations a model should be robust to (Morris et al., 2020) and strong attacks incorporate such constraints (Gao et al., 2018; Ebrahimi et al., 2018; Liu et al., 2022; Abad Rocamora et al., 2024). Despite the success of verification methods in the text domain, existing methods can only certify probabilistically via randomized smoothing (Cohen et al., 2019; Ye et al., 2020; Huang et al., 2023), or can only handle specifications such as replacements of characters/words, stop-word removal or word duplication (Huang et al., 2019; Jia et al., 2019; Shi et al., 2020; Zhang et al., 2021).

On the performance side, most successful certification methods rely on Interval Bound Propagation (IBP) (Moore et al., 2009), which in the text domain requires multiple forward passes through the first layers of the model (Huang et al., 2019), unlike in the image domain where a single forward pass is enough for verification (Wang et al., 2018). Moreover, IBP has been shown to provide a suboptimal verified accuracy in the image domain (Wang et al., 2021).

In the image domain, a popular approach to get fast robustness certificates is computing upper bounds on the Lipschitz constant of classifiers, and using this information to directly verify with a single forward pass (Hein and Andriushchenko, 2017; Tsuzuku et al., 2018; Latorre et al., 2020; Xu et al., 2022). These methods cannot be trivially applied in NLP because they assume the input to be in an $\ell_p$ space such $\mathbb{R}^d$, which is not the case of text input, where the input length can vary and inputs are discrete (characters). Therefore, we need to rethink Lipschitz verification for NLP.

In this work, we introduce the first method able to provide deterministic Levenshtein distance certificates for convolutional classifiers. This is achieved by computing the Lipschitz constant of intermediate layers with respect to the ERP distance (Chen and Ng, 2004). Our Lipschitz constant estimates

Table 1: **State of the art in Levenstein distance verification and our contributions:** `LipsLev` is the first to verify deterministically against Levenshtein distance constraints in a single forward pass.

| Method | Insertions/deletions | Deterministic | Single forward pass |
|---|:---:|:---:|:---:|
| Huang et al. (2019) | ✗ | ✓ | ✗ |
| Huang et al. (2023) | ✓ | ✗ | ✗ |
| `LipsLev` (Ours) | ✓ | ✓ | ✓ |

allow enforcing 1-Lipschitzness during training in order to achieve a larger verified accuracy. Our experiments in the AG-News, SST-2, Fake-News and IMDB datasets show non-trivial certificates at distances 1 and 2, taking 4 to 7 orders of magnitudes less time to verify. Furthermore, our method is the only one able to verify under Levenshtein distance larger than 1. We set the foundations for Lipschitz verification in NLP and we believe our method can be extended to more complex models.

**Notation:** We use uppercase bold letters for matrices $\boldsymbol{X} \in \mathbb{R}^{m \times n}$, lowercase bold letters for vectors $\boldsymbol{x} \in \mathbb{R}^m$ and lowercase letters for numbers $x \in \mathbb{R}$. Accordingly, the $i^{\text{th}}$ row and the element in the $i, j$ position of a matrix $\boldsymbol{X}$ are given by $\boldsymbol{x}_i$ and $x_{ij}$ respectively. We use the shorthand $[n] = \{0, 1, \cdots, n-1\}$ for any natural number $n$. Given two matrices $\boldsymbol{A} \in \mathbb{R}^{m \times d}$ and $\boldsymbol{B} \in \mathbb{R}^{n \times d}$ $\boldsymbol{A} \oplus \boldsymbol{B} = \begin{bmatrix} \boldsymbol{A} \\ \boldsymbol{B} \end{bmatrix} \in \mathbb{R}^{(m+n) \times d}$. Concatenating with the empty sequence $\emptyset$ results in the identity $\boldsymbol{A} \oplus \emptyset = \boldsymbol{A}$. We denote as $\boldsymbol{A}_{2:} \in \mathbb{R}^{(m-1) \times d}$ the matrix obtained by removing the first row. We denote the zero vector as $\boldsymbol{0}$ with dimensions appropriate to context. We use the operator $|\cdot|$ for the size of sets, e.g., $|\mathcal{S}(\Gamma)|$ and the length of sequences, e.g., for $\boldsymbol{X} \in \mathbb{R}^{m \times n}$, we have $|\boldsymbol{X}| = m$.

## 2 RELATED WORK

**Lipschitz verification:** Hein and Andriushchenko (2017) are the first to study the computation of the Lipschitz constant in order to provide formal guarantees of the robustness of support vector machines and two-layer nueral networks. Tsuzuku et al. (2018) compute Lipschitz constant upper bounds for deeper networks and regularize such upper bounds to improve certificates. Since then, tighter upper bounds for the Lipschitz constant have been proposed (Huang et al., 2021; Fazlyab et al., 2019; Latorre et al., 2020; Shi et al., 2022). A variety of works propose constraining the Lipschitz constant to be 1 during training in order to have automatic robustness certificates (Cisse et al., 2017; Qian and Wegman, 2019; Gouk et al., 2021; Xu et al., 2022). All previous works center in the standard $\ell_p$ norms and cannot be applied to the NLP domain. Our work provides the first 1-Lipschitz training method for the Levenshtein distance.

**Verfication in NLP:** Jia et al. (2019) propose using Interval Bound Propagation via an over-approximation of the embeddings of the set of synonyms of each word. Concurrently, Huang et al. (2019) incorporate this technique for verifying against replacements of nearby characters in the English keyboard. Bonaert et al. (2021); Shi et al. (2020) propose zonotope abstractions and IBP for verifying against synonym substitutions in transformer models. Zhang et al. (2021) propose a verification procedure that can handle a small number of input perturbations for LSTM classifiers. Deviating from these approaches, Ye et al. (2020) propose using randomized smoothing techniques Cohen et al. (2019) in order to verify probabilistically against character substitutions. Huang et al. (2023) used similar techniques in order to probabilistically verify under Levenshtein distance specifications. Zhao et al. (2022) propose a framework to verify under word substitutions via Causal Interventions. Sun and Ruan (2023) derive probable upper and lower bounds of the certified radius under word substitutions. Zhang et al. (2024) employ randomized smoothing to verify against word (synonym) substitutions, insertions, deletions and reorderings. Zeng et al. (2023) propose a randomized smoothing technique that does not rely on knowing how attackers generate synonyms. In Table 1 we highlight the differences with existing works in NLP verification.

## 3 PRELIMINARIES

Let $\mathcal{S}(\Gamma) = \{c_1 c_2 \cdots c_m : c_i \in \Gamma \ \forall m \in \mathbb{N} \setminus 0\}$ be the space of sequences of characters in the alphabet set $\Gamma$. We represent sentences $\boldsymbol{S} \in \mathcal{S}(\Gamma)$ as sequences of one-hot vectors, i.e., $\boldsymbol{S} \in$

$\{0,1\}^{m \times |\Gamma|} : ||s_i||_1 = 1, \; \forall i \in [m]$. Given a classification model $\boldsymbol{f} : \mathcal{S}(\Gamma) \to \mathbb{R}^o$ assigning scores to each of the $o$ classes, the predicted class for some $\boldsymbol{S} \in \mathcal{S}(\Gamma)$ is given by $\hat{y} = \arg\max_{i \in [o]} f(\boldsymbol{S})_i$. Our goal is to check whether for a given pair $(\boldsymbol{S}, y) \in (\mathcal{S}(\Gamma) \times [o])$:

$$f(\boldsymbol{S}')_y - \max_{\hat{y} \neq y} f(\boldsymbol{S}')_{\hat{y}} > 0, \; \forall \boldsymbol{S}' \in \mathcal{S}(\Gamma) : d_{\mathrm{Lev}}(\boldsymbol{S}, \boldsymbol{S}') \leq k, \tag{1}$$

where $d_{\mathrm{Lev}}$ is the Levenshtein distance (Levenshtein et al., 1966). The Levenshtein distance is defined as follows:

$$d_{\mathrm{Lev}}(\boldsymbol{S}, \boldsymbol{S}') := \begin{cases} |\boldsymbol{S}| & \text{if } |\boldsymbol{S}'| = 0 \\ |\boldsymbol{S}'| & \text{if } |\boldsymbol{S}| = 0 \\ d_{\mathrm{Lev}}(\boldsymbol{S}_{2:}, \boldsymbol{S}'_{2:}) & \text{if } s_1 = s'_1 \\ 1 + \min \begin{Bmatrix} d_{\mathrm{Lev}}(\boldsymbol{S}_{2:}, \boldsymbol{S}'_{2:}) \\ d_{\mathrm{Lev}}(\boldsymbol{S}_{2:}, \boldsymbol{S}') \\ d_{\mathrm{Lev}}(\boldsymbol{S}, \boldsymbol{S}'_{2:}) \end{Bmatrix} & \text{otherwise .} \end{cases}$$

The Levenshtein distance captures the number of character replacements, insertions or deletions needed in order to transform $\boldsymbol{S}$ into $\boldsymbol{S}'$ and vice-versa. Such constraints are employed in popular NLP attacks in order to enforce the imperceptibility of the attack (Gao et al., 2018; Ebrahimi et al., 2018; Liu et al., 2022; Abad Rocamora et al., 2024) following the findings of Morris et al. (2020).

## 3.1 Interval Bound Propagation (IBP)

Existing robustness verification approaches rely on IBP for verifying the robustness of text models (Huang et al., 2019; Jia et al., 2019). IBP relies on the input being constrained in a box. Let $\boldsymbol{x}, \boldsymbol{l}, \boldsymbol{u} \in \mathbb{R}^d$, every element of $\boldsymbol{x}$ is assumed to be in an interval given by $\boldsymbol{l}$ and $\boldsymbol{u}$, i.e., $l_i \leq x_i \leq u_i \; \forall i \in [d]$ or $\boldsymbol{x} \in [\boldsymbol{l}, \boldsymbol{u}]$ for short. These constraints arise naturally when studying robustness in the $\ell_\infty$ norm, as the constraint $\boldsymbol{x} \in \{\boldsymbol{x}^{(0)} + \boldsymbol{\delta} : ||\boldsymbol{\delta}||_\infty \leq \epsilon\}$ can exactly be represented as $\boldsymbol{x} \in [\boldsymbol{x}^{(0)} - \epsilon, \boldsymbol{x}^{(0)} + \epsilon]$. IBP consists in a set of rules to obtain interval constraints of the output of a function, given the interval constraints of the input. In the case of an affine mapping $\boldsymbol{f}(\boldsymbol{x}) = \boldsymbol{W}\boldsymbol{x} + \boldsymbol{b}$, we can easily obtain the interval constraints $\boldsymbol{f}(\boldsymbol{x}) \in [\boldsymbol{l_f}(\boldsymbol{x}), \boldsymbol{u_f}(\boldsymbol{x})], \; \forall \boldsymbol{x} \in [\boldsymbol{l}, \boldsymbol{u}]$ with:

$$\boldsymbol{l_f}(\boldsymbol{x}) = \boldsymbol{W}^+ \boldsymbol{l} + \boldsymbol{W}^- \boldsymbol{u} + \boldsymbol{b}, \quad \boldsymbol{u_f}(\boldsymbol{x}) = \boldsymbol{W}^+ \boldsymbol{u} + \boldsymbol{W}^- \boldsymbol{l} + \boldsymbol{b}, \tag{2}$$

where $\boldsymbol{W}^+$ and $\boldsymbol{W}^-$ are the positive and negative parts of $\boldsymbol{W}$. In the case of the ReLU activation function $\boldsymbol{\sigma}(\boldsymbol{x}) = \max\{0, \boldsymbol{x}\}$, we have that:

$$\boldsymbol{l_\sigma}(\boldsymbol{x}) = \boldsymbol{\sigma}(\boldsymbol{l}), \quad \boldsymbol{u_\sigma}(\boldsymbol{x}) = \boldsymbol{\sigma}(\boldsymbol{u}). \tag{3}$$

By applying recursively the simple rules in Eqs. (2) and (3), one can easily verify robustness properties of ReLU fully-connected and convolutional networks (Wang et al., 2018).

Nevertheless, IBP has two main limitations:

    a) IBP assumes the input space to be of fixed length, e.g., $\mathbb{R}^d$.

    b) IBP can only handle interval constrained inputs, e.g., $\boldsymbol{x} \in [\boldsymbol{l}, \boldsymbol{u}]$.

Limitation a) makes it impossible to verify Levenshtein distance constraints as they include insertion and deletion operations, which change the length of the input sequence. In the literature, limitation a) forces existing verification methods to only consider replacements of characters/words (Huang et al., 2019; Jia et al., 2019; Shi et al., 2020; Bonaert et al., 2021; Zhang et al., 2021).

Limitation b) can be circumvented by building an over approximation of the replacement constraints that can be represented with intervals. In the case of text, one can directly build an over approximation of the embeddings. Let $\boldsymbol{Z} = \boldsymbol{S}\boldsymbol{E} \in \mathbb{R}^{m \times d}$, where $\boldsymbol{S} \in \mathcal{S}(\Gamma)$ is the sequence of one-hot vectors representing each character/word, and $\boldsymbol{E} \in \mathbb{R}^{|\Gamma| \times d}$ is the embedding matrix. Let $d_{\mathrm{edit}}$ be the edit distance without insertions and deletions, our constraint in the edit distance (Eq. (1)) translates in the embedding space to the set:

$$\mathcal{Z}_k(\boldsymbol{S}) = \{\boldsymbol{S}'\boldsymbol{E} : d_{\mathrm{edit}}(\boldsymbol{S}, \boldsymbol{S}') \leq k, \boldsymbol{S}' \in \mathcal{S}(\Gamma)\},$$

where $d_{\mathrm{edit}}(\mathbf{S}, \mathbf{S}') = \sum_{i=1}^m ||s_i - s'_i||_\infty$ for any length $m$ sequences of one-hot vectors $\boldsymbol{S}, \boldsymbol{S}'$. We can overapproximate this set with interval constraints such that $\hat{\boldsymbol{Z}} \in [\boldsymbol{L}, \boldsymbol{U}]$, with $l_{ij} = \min_{\boldsymbol{Z} \in \mathcal{Z}_k(\boldsymbol{S})} z_{ij}$

and $u_{ij} = \max_{\boldsymbol{Z} \in \mathcal{Z}_k(\boldsymbol{S})} z_{ij}$. But, because we can replace any character/word at any position, we end up with $\boldsymbol{L} = \boldsymbol{l} \oplus \boldsymbol{l} \oplus \cdots \oplus \boldsymbol{l}$ and $\boldsymbol{U} = \boldsymbol{u} \oplus \boldsymbol{u} \oplus \cdots \oplus \boldsymbol{u}$, where:

$$l_i = \min_{k \in [|\Gamma|]} e_{ki}, \quad u_i = \max_{k \in [|\Gamma|]} e_{ki}, \quad \forall i \in [d].$$

Therefore, this overapproximation contains the embeddings of any $\boldsymbol{S}' \in \{0,1\}^{m \times |\Gamma|} : ||\boldsymbol{s}'_i||_1 = 1, \; \forall i \in [m]$, i.e., *every sentence of length $m$*, making verification impossible. To circumvent this, existing methods focus on the synonym replacement task, further restricting $\mathcal{Z}_k(\boldsymbol{S})$ to only replace words for a word in a small set of synonyms (Jia et al., 2019; Shi et al., 2020; Bonaert et al., 2021). Alternatively, Huang et al. (2019) compute the over approximation after the pooling layer of the model, circumventing this problem. Nevertheless, their approach requires $|\mathcal{Z}_1(\boldsymbol{S})|$ forward passes. This number of forward passes can be in the order of tens of thousands for large $m$ and $|\Gamma|$.

Our Lipschitz constant based approach, `LipsLev`, can handle sequences of any length and requires a single forward pass through the model.

## 4 METHOD

In Section 4.1 we cover the verification procedure once the Lipschitz constant of a classifier is known. In Section 4.2 we cover the convolutional architectures employed in Huang et al. (2019) and our Lipschitz constant estimation for them. Lastly, we introduce our training strategy in order to achieve non-trivial verified accuracy in Section 4.3. We defer our proofs to Appendix B.

### 4.1 LIPSCHITZ CONSTANT BASED VERIFICATION

Motivated by the success and efficiency of Lipschitz constant based certification in vision tasks (Huang et al., 2021; Xu et al., 2022), we propose a method of this kind that can handle previously studied models in the character-level classification task (Huang et al., 2019), and provide Levenshtein distance certificates.

Our goal is to compute the global Lipschitz constant. Let $g_{y,\hat{y}}(\boldsymbol{S}) = f(\boldsymbol{S})_y - f(\boldsymbol{S})_{\hat{y}}$ be the margin function for classes $y$ and $\hat{y}$, we would like to have for some $\boldsymbol{S}$:

$$|g_{y,\hat{y}}(\boldsymbol{S}) - g_{y,\hat{y}}(\boldsymbol{S}')| \leq G_{y,\hat{y}} \cdot d_{\text{Lev}}(\boldsymbol{S}, \boldsymbol{S}') \; \forall \boldsymbol{S}' \in \mathcal{S}(\Gamma), \tag{4}$$

for some $G_{y,\hat{y}} \in \mathbb{R}^+$. Given Eq. (4) is satisfied, the maximum distance up to which we can verify Eq. (1), is lower bounded by:

$$\max\{k : g_{y,\hat{y}}(\boldsymbol{S}') > 0 \; \forall \boldsymbol{S}' \in \mathcal{S}(\Gamma) : d_{\text{Lev}}(\boldsymbol{S}, \boldsymbol{S}') \leq k\} \geq \left\lfloor \frac{g_{y,\hat{y}}(\boldsymbol{S})}{G_{y,\hat{y}}} \right\rfloor. \tag{5}$$

Let $k^\star_{y,\hat{y}}(\boldsymbol{S}) := \left\lfloor \frac{g_{y,\hat{y}}(\boldsymbol{S})}{G_{y,\hat{y}}} \right\rfloor$, we denote $k^\star_y(\boldsymbol{S}) := \min_{\hat{y} \neq y} k^\star_{y,\hat{y}}(\boldsymbol{S})$ to be the *certified radius*.

### 4.2 LIPSCHITZ CONSTANT ESTIMATION FOR CONVOLUTIONAL CLASSIFIERS

Let $\boldsymbol{S} \in \mathcal{S}(\Gamma)$ be a sequence of one-hot vectors, our classifier is defined as:

$$\boldsymbol{f}(\boldsymbol{S}) = \left( \sum_{i=1}^{m+l \cdot (q-1)} f_i^{(l)}(\boldsymbol{S}) \right) \boldsymbol{W}, \text{ where } \boldsymbol{f}^{(j)}(\boldsymbol{S}) = \begin{cases} \boldsymbol{\sigma} \left( \boldsymbol{C}^{(j)} \left( \boldsymbol{f}^{(j-1)}(\boldsymbol{S}) \right) \right) & \forall j = 1, \cdots, l \\ \boldsymbol{S}\boldsymbol{E} & j = 0 \end{cases}, \tag{6}$$

where $\boldsymbol{E} \in \mathbb{R}^{v \times d}$ is the embeddings matrix, $\boldsymbol{C}^{(i)}, \forall i = 1, \cdots, l$ are convolutional layers with kernel size $q$ and hidden dimension $k$. $\sigma$ is the ReLU activation function and $\boldsymbol{W} \in \mathbb{R}^{k \times o}$ is the last classification layer. This architecture was previously studied in verification by (Huang et al., 2019; Jia et al., 2019).

Our approach to estimate the global Lipschitz constant of such a classifier is to compute the Lipschitz constant of each layer. Then, since the overall function in Eq. (6) is the sequential composition of all of the layers, we can just multiply the Lipschitz constants to obtain the global one. However, in order to be able to do this, we need some metric with respect to which we can compute the Lipschitz

constant. The Levenshtein distance cannot be applied, as it can only measure distances between one-hot vectors and the outputs of intermediate layers are sequences of real vectors. For this task, we select the *ERP distance* (Chen and Ng, 2004):

**Definition 4.1** (ERP distance (Chen and Ng, 2004)). Let $\boldsymbol{A} \in \mathbb{R}^{m \times d}$ and $\boldsymbol{B} \in \mathbb{R}^{n \times d}$ be two sequences of $m$ and $n$ real vectors respectively and $p \geq 1$. The ERP distance is defined as:

$$
d_{\text{ERP}}^p(\boldsymbol{A}, \boldsymbol{B}) = \begin{cases} \sum_{i=1}^{m} ||\boldsymbol{a}_i||_p & \text{if } n = 0 \ (\boldsymbol{B} = \emptyset) \\ \sum_{i=1}^{n} ||\boldsymbol{b}_i||_p & \text{if } m = 0 \ (\boldsymbol{A} = \emptyset) \\ \min \left\{ \begin{array}{l} ||\boldsymbol{a}_1||_p + d_{\text{ERP}}^p(\boldsymbol{A}_{2:}, \boldsymbol{B}), \\ ||\boldsymbol{b}_1||_p + d_{\text{ERP}}^p(\boldsymbol{A}, \boldsymbol{B}_{2:}), \\ ||\boldsymbol{a}_1 - \boldsymbol{b}_1||_p + d_{\text{ERP}}^p(\boldsymbol{A}_{2:}, \boldsymbol{B}_{2:}) \end{array} \right\} & \text{otherwise} \end{cases}
$$

The ERP distance is a natural extension of the Levenshtein distance for sequences of real valued vectors. In fact, in the case we compare sequences of one-hot vectors and we set $p = \infty$, we recover the Levenshtein distance, see Lemma S4.

In the following we define a useful representation of convolutional layers.

**Definition 4.2** (1D Convolutional layer with zero padding). Let $\boldsymbol{A} \in \mathbb{R}^{m \times d}$ be a sequence of $d$-dimensional vectors. Let $k$ be the number of filters and $q$ the kernel size, a convolutional layer $\boldsymbol{C} : \mathbb{R}^{m \times d} \to \mathbb{R}^{(m+q-1) \times k}$ with parameters $\mathcal{K} \in \mathbb{R}^{q \times k \times d}$ can be represented as:

$$
\boldsymbol{c}_i(\boldsymbol{A}) = \sum_{j=1}^{m+2 \cdot (q-1)} \hat{\boldsymbol{K}}_{i,j} \hat{\boldsymbol{a}}_j, \quad \text{where } \hat{\boldsymbol{K}}_{i,j} = \begin{cases} \boldsymbol{K}_{j-i+1} & \text{if } 0 \leq j - i \leq q - 1 \\ \boldsymbol{00}^\top & \text{otherwise} \end{cases}, \quad \forall i \in [m+q-1],
$$

and $\hat{\boldsymbol{A}} = \boldsymbol{0}_{(q-1) \times d} \oplus \boldsymbol{A} \oplus \boldsymbol{0}_{(q-1) \times d} \in \mathbb{R}^{(m+2 \cdot (q-1)) \times d}$ is the zero-padded input. We denote the parameter tensor corresponding to every layer $\boldsymbol{C}^{(i)}$ as $\mathcal{K}^{(i)}$.

In Theorem 4.3 we present our Lipschitz constant upper bound. In Corollary 4.4 the Lipschitz constant upper bound is employed to compute the certified radius at a sentence $\boldsymbol{P}$. The Lipschitz constant upper bound can be further refined considering the local Lipschitz constant of the embedding layer around sentence $\boldsymbol{P}$, see Remark 4.5.

**Theorem 4.3** (Lipschitz constant of margins of convolutional models). *Let $\boldsymbol{f}$ be defined as in Eq. (6). Let $g_{y,\hat{y}}(\boldsymbol{S}) = \boldsymbol{f}(\boldsymbol{S})_y - \boldsymbol{f}(\boldsymbol{S})_{\hat{y}}$ be the margin function for classes $y$ and $\hat{y}$. Let $p \geq 1$. Let $\boldsymbol{P}$ and $\boldsymbol{Q}$ be sequences of one-hot vectors, we have that for any $y$ and $\hat{y}$:*

$$
|g_{y,\hat{y}}(\boldsymbol{P}) - g_{y,\hat{y}}(\boldsymbol{Q})| \leq ||\boldsymbol{w}_{\hat{y}} - \boldsymbol{w}_y||_r \cdot \left( \prod_{i=1}^{l} M\left(\mathcal{K}^{(i)}\right) \right) \cdot M(\boldsymbol{E}) \cdot d_{Lev}(\boldsymbol{P}, \boldsymbol{Q}),
$$

*where $M(\mathcal{K}) = \sum_{i=1}^{q} ||\boldsymbol{K}_i||_p$, $M(\boldsymbol{E}) = \max\{\max_{i \in [|\Gamma|]} ||\boldsymbol{e}_i||_p, \max_{i,j \in [|\Gamma|]} ||\boldsymbol{e}_i - \boldsymbol{e}_j||_p\}$ and $\frac{1}{p} + \frac{1}{r} = 1$.*[1]

*Proof.* See Appendix B $\qquad \square$

**Corollary 4.4** (Certified radius of convolutional models). *Let $\boldsymbol{f}$ be defined as in Eq. (6) and the Lipschitz constant of $g_{y,\hat{y}}$ be:*

$$
G_{y,\hat{y}} = ||\boldsymbol{w}_{\hat{y}} - \boldsymbol{w}_y||_r \cdot \left( \prod_{i=1}^{l} M\left(\mathcal{K}^{(i)}\right) \right) \cdot M(\boldsymbol{E}).
$$

*Then, the certified radius of $\boldsymbol{f}$ at the sentence $\boldsymbol{P}$ is given by: $k_{y,\hat{y}}^\star(S) = \min_{\hat{y} \neq y} \left\lfloor \frac{g_{y,\hat{y}}(\boldsymbol{P})}{G_{y,\hat{y}}} \right\rfloor$.*

**Remark 4.5** (Local Lipschitz constant of the embedding layer). Let the embeddings of a sentence $\boldsymbol{S}$ be given by $\boldsymbol{SE}$, we have that for any two sentences $\boldsymbol{P}$ and $\boldsymbol{Q}$:

$$
d_{\text{ERP}}^p(\boldsymbol{PE}, \boldsymbol{QE}) \leq M(\boldsymbol{E}, \boldsymbol{P}) \cdot d_{\text{Lev}}(\boldsymbol{P}, \boldsymbol{Q}),
$$

where $M(\boldsymbol{E}, \boldsymbol{P}) = \max\{\max_{i \in [|\Gamma|]} ||\boldsymbol{e}_i||_p, \max_{i \in |\boldsymbol{P}|, j \in [d]} ||\boldsymbol{p}_i \boldsymbol{E} - \boldsymbol{e}_j||_p\}$, satisfying $M(\boldsymbol{E}, \boldsymbol{P}) \leq M(\boldsymbol{E})$.

---

[1] In the case $p = 1$ and $p = \infty$, we have $r = \infty$ and $r = 1$ respectively.

### 4.3 Training 1-Lipschitz classifiers

Models trained with the standard Cross Entropy loss and Stochastic Gradient Descent (SGD) recipe are not amenable to verification methods, resulting in small certified radiuses. This has motivated the use of specialized training methods in the image domain (Mirman et al., 2018; Gowal et al., 2018; Mueller et al., 2023; Palma et al., 2024). Verification methods in the text domain also require tailored training methods to achieve non-zero certified radiuses (Huang et al., 2019; Jia et al., 2019). Motivated by methods enforcing classifiers to be 1-Lipschitz in the image domain (Xu et al., 2022), we enforce this constraint during training in order to improve certification.

In order to achieve a 1-Lipschitz classifier, we enforce 1-Lipschitzness of every layer by dividing the output of each layer by its Lipschitz constant. This results in our modified classifier being:

$$\hat{\boldsymbol{f}}(\boldsymbol{S}) = \left(\sum_{i=1}^{m+l\cdot(q-1)} \hat{f}_i^{(l)}(\boldsymbol{S})\right) \frac{\boldsymbol{W}}{M(\boldsymbol{W})}, \text{ where } \hat{\boldsymbol{f}}^{(j)}(\boldsymbol{S}) = \begin{cases} \frac{\boldsymbol{\sigma}\left(\boldsymbol{C}^{(j)}\left(\hat{\boldsymbol{f}}^{(j-1)}(\boldsymbol{S})\right)\right)}{M(\boldsymbol{\mathcal{K}}^{(j)})} & \forall j = 1, \cdots, l \\ \frac{\boldsymbol{S}\boldsymbol{E}}{M(\boldsymbol{E})} & j = 0 \end{cases},$$
(7)

where $M(\boldsymbol{W}) = \max_{y,\hat{y}\in[o]} ||\boldsymbol{w}_y - \boldsymbol{w}_{\hat{y}}||_r$. Note that the last layer is made 1-Lipschitz with respect to the worst pair of class labels. Incorporating this information and Remark 4.5, we end up with the final Lipschitz constant for the classifier:

**Corollary 4.6** (Local Lipschitz constant of modified classifiers). *Let $\hat{f}$ be defined as in Eq. (7). Let $\hat{g}_{y,\hat{y}}(\boldsymbol{S}) = \hat{f}(\boldsymbol{S})_y - \hat{f}(\boldsymbol{S})_{\hat{y}}$ be the margin function for classes $y$ and $\hat{y}$. Let $\boldsymbol{P}$ and $\boldsymbol{Q}$ be sequences of one-hot vectors, we have that for any $y$ and $\hat{y}$:*

$$|\hat{g}_{y,\hat{y}}(\boldsymbol{P}) - \hat{g}_{y,\hat{y}}(\boldsymbol{Q})| \le \frac{||\boldsymbol{w}_{\hat{y}} - \boldsymbol{w}_y||_r}{M(\boldsymbol{W})} \cdot \frac{M(\boldsymbol{E}, \boldsymbol{P})}{M(\boldsymbol{E})} \cdot d_{lev}(\boldsymbol{P}, \boldsymbol{Q}),$$

*where $M(\boldsymbol{E})$ is defined as in Theorem 4.3, $M(\boldsymbol{E}, \boldsymbol{P})$ is as in Remark 4.5 and $M(\boldsymbol{W}) = \max_{y,\hat{y}\in[o]} ||\boldsymbol{w}_y - \boldsymbol{w}_{\hat{y}}||_r$. Note that this Lipschitz constant is local as it depends on $\boldsymbol{P}$.*

Note that the local Lipschitz constant estimate in Corollary 4.6 is guaranteed to be at most 1 as $||\boldsymbol{w}_{\hat{y}} - \boldsymbol{w}_y||_r \le M(\boldsymbol{W})$ and $M(\boldsymbol{E}, \boldsymbol{P}) \le M(\boldsymbol{E})$. Given this estimate, we can proceed similarly to Corollary 4.4 in order to obtain the certified radius of the modified model. Note that in the forward pass of Eq. (7), we need to compute $M(\boldsymbol{E}), M(\boldsymbol{\mathcal{K}}^{(j)})$ and $M(\boldsymbol{W})$, which increases the complexity of a forward pass with respect to Eq. (6). Nevertheless, we observe this can be efficiently done during training as seen in Appendix A.6. Then, the weights of each layer can be divided by its Lipschitz constant, resulting in the same architecture in Eq. (6) with the guarantees of Corollary 4.6.

## 5 Experiments

In this section, we cover our experimental validation. In Section 5.1 we cover the experimental setup and training mechanisms shared among all experiments. In Section 5.2 we compare performance of our approach with existing IBP approaches and the naive brute force verification baseline. Lastly, in Section 5.3 we cover the hyperparameter selection of our method. We define our performance metrics and perform additional experiments in Appendix A.

### 5.1 Experimental setup

We train and verify our models in the sentence classification datasets AG-News (Gulli, 2005; Zhang et al., 2015), SST-2 (Wang et al., 2019), IMDB (Maas et al., 2011) and Fake-News (Lifferth, 2018). We consider all of the characters present in the dataset except for uppercase letters, which we tokenize as lowercase. Each character is tokenized individually and assigned one embedding vector via the matrix $\boldsymbol{E}$. For all our models and datasets, following Huang et al. (2019), we train models with a single convolutional layer, an embedding size of 150, a hidden size of 100 and a kernel size of 5 for the SST-2 dataset and 10 for the rest of datasets. Following the setup used in Andriushchenko and Flammarion (2020) for adversarial training, we use the SGD optimizer with batch size 128 and a 30-epoch cyclic learning rate scheduler with a maximum value of 100.0, which we select via a grid search in a validation dataset, see Appendix A.5. For every experiment, we report the average results over three random seeds and report the performance over the first 1,000 samples of the test set. Our

standard deviations are reported in Appendix A.3. Due to the extreme time costs of the brute-force and IBP approaches in the Fake-News dataset, we reduce to 50 samples in this dataset. We compute the adversarial accuracy with Charmer (Abad Rocamora et al., 2024). For completeness, we report the performance of `LipsLev` over the full $1,000$ samples and $k$ up to 10 in Appendix A.4. All of our experiments are conducted in a single machine with an NVIDIA A100 SXM4 40 GB GPU. Our implementation is available in github.com/LIONS-EPFL/LipsLev.

Table 2: **Verified accuracy under bounded $d_{\mathbf{lev}}$:** We report the Clean accuracy (Acc.), Adversarial Accuracy (Adv. Acc.) with Charmer (Abad Rocamora et al., 2024), Verified accuracy (Ver.) and the average runtime in seconds (Time) for the brute-force approach (BruteF), IBP (Huang et al., 2019) and `LipsLev`. **OOT** means the experiment was Out Of Time. ✗ means the method does not support $d_{\mathrm{lev}} > 1$. Our method, `LipsLev`, is the only method able to provide non-trivial verified accuracies for any $k$ in a single forward pass.

| Dataset | $p$ | $k$ | Acc.(%) | Charmer Adv. Acc.(%) | Time(s) | BruteF Ver.(%) | Time(s) | IBP Ver.(%) | Time(s) | LipsLev Ver.(%) | Time(s) |
|---|---|---|---|---|---|---|---|---|---|---|---|
| AG-News | $\infty$ | 1 | 65.23 | 47.90 | 5.70 | 47.87 | 16.15 | 27.77 | 16.76 | 32.33 | 0.0015 |
| | | 2 | | 32.97 | 5.70 | **OOT** | | ✗ | | 11.60 | 0.0015 |
| | 1 | 1 | 69.63 | 54.47 | 5.43 | 54.43 | 15.33 | 18.93 | 17.56 | 34.50 | 0.00140 |
| | | 2 | | 37.77 | 5.43 | **OOT** | | ✗ | | 12.53 | 0.00140 |
| | 2 | 1 | 74.80 | 62.20 | 7.32 | 62.07 | 29.12 | 29.10 | 31.54 | 38.80 | 0.00970 |
| | | 2 | | 46.47 | 7.32 | **OOT** | | ✗ | | 13.93 | 0.00970 |
| SST-2 | $\infty$ | 1 | 63.95 | 39.68 | 1.84 | 39.68 | 2.27 | 33.94 | 2.88 | 14.68 | 0.00084 |
| | | 2 | | 19.92 | 1.84 | **OOT** | | ✗ | | 0.99 | 0.00084 |
| | 1 | 1 | 69.69 | 45.26 | 1.91 | 45.22 | 2.31 | 19.00 | 2.99 | 18.69 | 0.0022 |
| | | 2 | | 26.11 | 1.91 | **OOT** | | ✗ | | 1.83 | 0.0022 |
| | 2 | 1 | 69.95 | 48.81 | 2.09 | 48.78 | 4.23 | 16.06 | 5.22 | 14.57 | 0.0047 |
| | | 2 | | 30.70 | 2.09 | **OOT** | | ✗ | | 0.73 | 0.0047 |
| Fake-News | $\infty$ | 1 | 100.00 | 86.67 | 66.82 | 86.67 | 972.46 | 85.33 | 989.84 | 85.33 | 0.017 |
| | | 2 | | 76.00 | 66.82 | **OOT** | | ✗ | | 68.67 | 0.017 |
| | 1 | 1 | 98.00 | 92.00 | 67.11 | 92.00 | 978.94 | 91.33 | 990.32 | 91.33 | 0.014 |
| | | 2 | | 79.33 | 67.11 | **OOT** | | ✗ | | 75.33 | 0.014 |
| | 2 | 1 | 98.00 | 88.67 | 73.52 | 88.67 | 1224.45 | 87.33 | 1466.38 | 87.33 | 0.0089 |
| | | 2 | | 78.00 | 73.52 | **OOT** | | ✗ | | 71.33 | 0.0089 |
| IMDB | $\infty$ | 1 | 74.57 | 67.43 | 14.16 | 67.43 | 130.49 | 61.50 | 138.20 | 31.37 | 0.0047 |
| | | 2 | | 59.77 | 14.16 | **OOT** | | ✗ | | 5.90 | 0.0047 |
| | 1 | 1 | 69.57 | 61.17 | 14.44 | 61.00 | 134.23 | 47.30 | 135.22 | 28.73 | 0.0027 |
| | | 2 | | 52.20 | 14.44 | **OOT** | | ✗ | | 6.80 | 0.0027 |
| | 2 | 1 | 60.60 | 46.87 | 16.24 | 46.73 | 261.99 | 37.57 | 308.73 | 8.67 | 0.0019 |
| | | 2 | | 35.10 | 16.24 | **OOT** | | ✗ | | 0.87 | 0.0019 |

## 5.2 COMPARISON WITH IBP AND BRUTE FORCE APPROACHES

In this section, we compare our verification method against a brute-force approach and a modification of the IBP method in (Huang et al., 2019) to handle insertions and deletions of characters.

With the brute-force approach, for every sentence $\boldsymbol{P}$ in the test dataset, we evaluate our model in every sentence in the set $\{\boldsymbol{Q} : d_{\mathrm{lev}}(\boldsymbol{P}, \boldsymbol{Q}) \leq k\}$ and check if there is any missclassification. Since the size of this set grows exponentially with $k$, we only evaluate the brute-force accuracy for $k = 1$.

In the case of IBP, we evaluate the classifier up to the pooling layer in every sentence of $\{\boldsymbol{Q} : d_{\mathrm{lev}}(\boldsymbol{P}, \boldsymbol{Q}) \leq k\}$ and then build the overapproximation. In (Huang et al., 2019) it was enough to build this overapproximation for $k = 1$ and re-scale it to capture larger $k$s. This is not the case for insertions and deletions, this constrains IBP with Levenshtein distance specifications to work only for $k = 1$. Overall, the complexity of IBP is the same as the brute-force approach without providing the exact robust accuracy. Because Huang et al. (2019) only considered perturbations of characters nearby in the English keyboard, the maximum perturbation size at $k = 1$ was very small, e.g., 206 and 722 sentences for SST-2 and AG-News respectively[2]. In our setup, the maximum perturbation sizes are $33,742$ and $85,686$. This makes it impractical to perform IBP verified training. We train 3 models for each dataset and $p \in \{1, 2, \infty\}$ and verify them with the three methods. We report the average time to verify and the clean, adversarial and verified accuracies at $k \in \{1, 2\}$.

---

[2]See Table 3 in Huang et al. (2019)

Table 3: **Regularizing v.s. enforcing Lipschitzness in SST-2:** We compare the performance when regularizing the Lipschitz constant ($G$) during training with $\lambda \in \{0, 0.001, 0.01, 0.1\}$, against enforcing 1-Lipschitzness through Eq. (7). Regularizing $G$ leads to either models with similar performance to a constant classifier (55.7% for SST-2), or more accurate but non-verifiable models than when using the formulation in Eq. (7).

| $\lambda$ | $p = \infty$ | | | $p = 1$ | | | $p = 2$ | | |
|---|---|---|---|---|---|---|---|---|---|
| | Clean(%) | Ver.(%) | $G$ | Clean(%) | Ver.(%) | $G$ | Clean(%) | Ver.(%) | $G$ |
| 0 | $89.0_{\pm(0.5)}$ | $0.0_{\pm(0.0)}$ | $2850.2_{\pm(80.1)}$ | $86.1_{\pm(0.4)}$ | $0.0_{\pm(0.0)}$ | $449.6_{\pm(3.0)}$ | $87.2_{\pm(0.2)}$ | $0.0_{\pm(0.0)}$ | $129.1_{\pm(2.9)}$ |
| 0.001 | $80.8_{\pm(0.6)}$ | $0.0_{\pm(0.0)}$ | $65.0_{\pm(0.9)}$ | $84.5_{\pm(0.5)}$ | $0.0_{\pm(0.0)}$ | $44.1_{\pm(0.7)}$ | $86.2_{\pm(0.4)}$ | $0.0_{\pm(0.0)}$ | $37.7_{\pm(0.3)}$ |
| 0.01 | $60.1_{\pm(1.1)}$ | $1.7_{\pm(0.1)}$ | $1.4_{\pm(0.1)}$ | $79.7_{\pm(0.5)}$ | $0.1_{\pm(0.0)}$ | $6.9_{\pm(0.0)}$ | $81.6_{\pm(0.4)}$ | $0.1_{\pm(0.0)}$ | $8.8_{\pm(0.1)}$ |
| 0.1 | $56.2_{\pm(0.0)}$ | $55.7_{\pm(0.3)}$ | $0.0_{\pm(0.0)}$ | $57.3_{\pm(0.0)}$ | $53.2_{\pm(0.8)}$ | $0.1_{\pm(0.0)}$ | $57.5_{\pm(0.9)}$ | $34.1_{\pm(3.0)}$ | $0.1_{\pm(0.0)}$ |
| Eq. (7) | $62.8_{\pm(0.6)}$ | $7.3_{\pm(0.1)}$ | $1.00_{\pm(0.0)}$ | $65.6_{\pm(0.1)}$ | $10.7_{\pm(0.2)}$ | $1.00_{\pm(0.0)}$ | $66.6_{\pm(0.6)}$ | $7.2_{\pm(0.1)}$ | $1.00_{\pm(0.0)}$ |

In Table 2, we can observe that the $p$ value has a big influence in the clean accuracy of the models and the verification capability of each method. With $p = 2$, we observe the highest clean accuracy AG-News and SST-2, with an average of 74.80% and 69.95% respectively. In the case of Fake-News and IMDB, $p = \infty$ provided the best accuracy with 100% and 74.57% respectively. In terms of robust accuracy (BruteF), $p = 2$ also provides the best performance with 62.07% for AG-News and 48.78% for SST-2, while for Fake-News and IMDB, the best performance was achieved with $p = 1$ and $p = \infty$ respectively (92% and 74.57%). We observe that IBP obtains the best ratio between clean and verified accuracy when employing $p = \infty$, providing the best performance in the IMDB and SST-2 datasets at $k = 1$. Our method, `LipsLev`, is able to improve over IBP in AG-News and match IBP in Fake-News at $k = 1$, being the only method able to verify for $k > 1$. At distance $k = 2$, we can observe that the Charmer adversarial accuracy in AG-News, SST-2 and IMDB is significantly larger than the verified accuracy given by `LipsLev`. Contrarily, for the Fake-News dataset, `LipsLev` is able to have a gap as close as 75.33% v.s. 79.33% with $p = 1$.

In terms of runtime, our method is from 4 to 7 orders of magnitude faster than brute-force and IBP, which attain similar runtimes. The impossibility of IBP to verify for $k > 1$ and its larger runtime than brute-force, poses it as an impractical tool for Levenstein distance verification. Our method is the only one able to verify for $k > 1$, with 13.93% verified accuracy for AG-News, 1.83% for SST-2, 75.33% for Fake-News and 6.80% for IMDB at $k = 2$.

## 5.3 REGULARIZING THE LIPSCHITZ CONSTANT

In Section 4.3 we describe how to enforce our convolutional classifier to be 1-Lipschitz. But, is there a better way of improving the final verified accuracy of our models? Because our Lipschitz constant estimate in Theorem 4.3 its differentiable with respect to the parameters of the model, we can regularize this quantity during training in order to achieve a lower Lipschitz constant and hopefully a better verified accuracy. In practice we regularize $G = M(\boldsymbol{W}) \cdot M(\boldsymbol{\mathcal{K}}^{(1)}) \cdot M(\boldsymbol{E})$ as defined in Theorem 4.3 and Corollary 4.6.

We train single-layer models with a regularization parameter of $\lambda \in \{0, 0.001, 0.01, 0.1\}$, where $\lambda = 0$ is equivalent to standard training. We initialize the weights of each layer so that their Lipschitz constant is 1. We use a learning rate of 0.01. We measure the final Lipschitz constant of each model and their clean and verified accuracies in a validation set of $1,000$ samples left out from the training set. As a baseline, we report these metrics for the models trained with the formulation in Eq. (7).

In Table 3 we observe that for all the studied norms, when regularizing the Lipschitz constant $G$, we cannot easily match the performance when using Eq. (7). Regularized models converge to either close-to-constant classifiers (55.7% clean accuracy for SST-2) or present a close-to-zero verified accuracy, This behavior has also been observed practically and theoretically for $\ell_p$ spaces (Zhang et al., 2022). The formulation in Eq. (7) allows us to obtain verifiable models without the need to tune hyperparameters.

## 5.4 THE INFLUENCE OF SENTENCE LENGTH IN VERIFICATION

In this section we study the qualitative characteristics of a sentence leading to better verification properties, specifically, we study the influence of the sentence length in verification. We compute

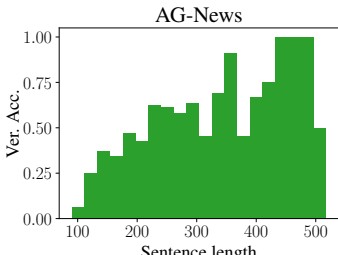 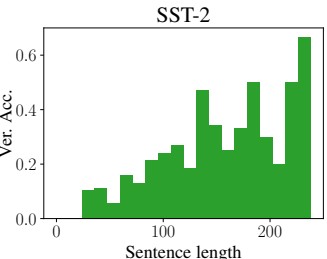

| Average Lentgh | | | |
|---|---|---|---|
| | | Verified | |
| Dataset | Method | ✗ | ✓ |
| AG-News | BruteF | 221.4 | 248.4 |
| | LipsLev | 228.0 | 258.9 |
| SST-2 | BruteF | 93.2 | 109.5 |
| | LipsLev | 98.3 | 126.8 |

Figure 1: **Sentence length distribution for verified and not verified sentences:** We report the verified accuracy v.s. sentence length with `LipsLev` (*Left*), and the average length of verified and not verified sentences (*Right*) at $k = 1$ in the models trained with $p = 2$. Shorter sentences are harder to verify in both SST-2 and AG-News with both `LipsLev` and the brute-force approach.

the verified accuracy v.s. sentence length at $k = 1$ for the models in Section 5.2 with `LipsLev` and $p = 2$. For the AG-News we remove the outlier sentences with length larger than 600 characters. The full length distribution is displayed in Appendix A.

In Fig. 1 we can observe that for both verification methods on both datasets, the verified sentences present a larger average length. Additionally, there is a clear increasing tendency in the verified accuracy v.s. sentence length. We believe this is reasonable as single characters perturbations are likely to introduce a smaller semantic change for longer sequences.

## 6 CONCLUSION

In this work, we propose the first approach able to verify NLP classifiers using the Levenshtein distance constraints. Our approach is based on an upper bound of the Lipschitz constant of convolutional classifiers with respect to the Levenshtein distance. Our method, `LipsLev` is able to obtain verified accuracies at any distance $k$ with single forward pass per sample. Moreover, our method is the only existing method that can practically verify for Levenshtein distances larger than $k = 1$. We expect our work can inspire a new line of works on verifying larger distances and more broadly verifying additional classes of NLP classifiers. We will make the code publicly available upon the publication of this work, our implementation is attached with this submission.

**Future directions and challenges:** A problem shared with verification methods in the image domain is scalability (Wang et al., 2021). Scaling verification methods to production models is a challenge, that becomes more relevant with the deployment of Large Language Models and their recently discovered vulnerabilities (Zou et al., 2023). Even though our method is the first to practically provide Levenshtein distance certificates in NLP, neither the formulation of Huang et al. (2019) or our formulation covers modern architectures as Transformers (Vaswani et al., 2017). We highlight the main challenges as follows:

i) **Tokenizers:** Modern Transformer-based classifiers utilize popular tokenizers such as SentencePiece (Kudo and Richardson, 2018), which aggregate contiguous characters in tokens before feeding them to the model. In order to deal with such non-differentiable piece, methods for computing the Lipschitz constant of tokenizers are needed.

ii) **Poor performance on character-level tasks:** In the case no tokenizer is used, transformers are known to fail in character-level classification tasks like the IMDB classification problem of Long Range Arena (Tay et al., 2021).

iii) **Non-Lipschitzness of Transformers:** Transformers are known to have a non-bounded Lipschitz constant (Kim et al., 2021). In the image domain, verification methods modify the model to be Lipschitz (Qi et al., 2023; Bonaert et al., 2021) or compute local Lipschitz constants (Havens et al., 2024). Nevertheless, it is non-trivial to extend such approaches from $\ell_p$-induced distances to the Levenshtein distance.

Our work sets the mathematical foundations of Lipschitz verification in NLP, opens the door to addressing these challenges and to achieving verifiable architectures beyond convolutional models.

ACKNOWLEDGEMENTS

Elias is thankful to Adrian Luis Müller and Ioannis Mavrothalassitis for the amazing discussions in the whiteboard of the Lab. Authors acknowledge the constructive feedback of reviewers and the work of ICLR'25 program and area chairs. We thank Zulip for their project organization tool. ARO - Research was sponsored by the Army Research Office and was accomplished under Grant Number W911NF-24-1-0048. Hasler AI - This work was supported by Hasler Foundation Program: Hasler Responsible AI (project number 21043). SNF project – Deep Optimisation - This work was supported by the Swiss National Science Foundation (SNSF) under grant number 200021_205011.

BROADER IMPACT

In this work, we tackle the important problem of verifying the robustness of NLP models against adversarial attacks. By advancing in this area, we can positively impact society by ensuring NLP models deployed in safety critical applications are robust to such perturbations.

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

## A ADDITIONAL EXPERIMENTAL VALIDATION

In Appendix A.1 we present the definition of the performance metrics employed in this work. In Appendix A.5 we present our grid search for selecting the best learning rate for each dataset and $p$ value in the ERP distance Definition 4.1. In Appendix A.6 we evaluate the effect of training with a different number of layers and report their latencies.

### A.1 DEFINITION OF PERFORMANCE METRICS

In this section we define the key metrics used to evaluate our models and verification methods. Let $\mathbb{1}\,()$ be the indicator function, given a classification model $\boldsymbol{f} : \mathcal{S}(\Gamma) \to \mathbb{R}^o$ assigning scores to each of the $o$ classes and a dataset $\mathcal{D} = \{\boldsymbol{S}^{(i)}, y^{(i)}\}_{i=1}^n$ with $\boldsymbol{S}^{(i)} \in \mathcal{S}(\Gamma)$ and $y^{(i)} \in [o]$, the clean, adversarial and verified accuracy are defined as:

**Definition S1** (Clean accuracy). The clean accuracy is a percentage in $[0, 100]$ that is computed as:

$$\text{Acc.}(\boldsymbol{f}, \mathcal{D}) \;=\; \frac{100}{n} \sum_{i=1}^n \mathbb{1}\left(y^{(i)} = \arg\max_{j \in [o]} f(\boldsymbol{S}^{(i)})_j\right).$$

**Definition S2** (Adversarial accuracy). Given an adversary $\boldsymbol{A} : \mathcal{S}(\Gamma) \to \mathcal{S}(\Gamma)$ that perturbs a sentence[3]. The adversarial accuracy is a percentage in $[0, 100]$ that is computed as:

$$\text{Adv. Acc.}(\boldsymbol{f}, \mathcal{D}, \boldsymbol{A}) \;=\; \frac{100}{n} \sum_{i=1}^n \mathbb{1}\left(y^{(i)} = \arg\max_{j \in [o]} f(\boldsymbol{A}(\boldsymbol{S}^{(i)}))_j\right).$$

**Definition S3** (Verified accuracy). Given a verification method $v$ returning the certified radius (see Section 4.1) for a given model $\boldsymbol{f}$ and sample $(\boldsymbol{S}, y)$ as $v(\boldsymbol{f}, \boldsymbol{S}, y) \in \{0\} \cup \mathbb{N}$. The verified accuracy at distance $k$ is a percentage in $[0, 100]$ that is computed as:

$$\text{Ver. Acc.}(\boldsymbol{f}, \mathcal{D}, v, k) \;=\; \frac{100}{n} \sum_{i=1}^n \mathbb{1}\left(v(\boldsymbol{f}, \boldsymbol{S}^{(i)}, y^{(i)}) \geq k\right).$$

For simplicity, the arguments of each accuracy function are omitted in the text as they can be inferred from the context.

### A.2 SENTENCE LENGTH DISTRIBUTION

In this section, we provide additional details about the sequence length distribution of verified and not verified sentences. Specifically, in Fig. S2 we provide the full distribution of lengths from the experiment in Fig. 1.

### A.3 STANDARD DEVIATIONS

In Table S4 we report the results from Table 2 with standard deviations for completeness.

### A.4 LARGER DISTANCES FOR FAKE-NEWS

In Section 5.2 we reported the performance in the Fake-News dataset over the first 50 samples of the test set and only up to $k = 2$. Nevertheless, the speed of `LipsLev` allows for more samples and larger distances. In Table S5, we evaluate the performance of `LipsLev` over the first $1,000$ test samples and up to $k = 10$.

---

[3] The adversary will usually adhere to some constraints such as $d_{\text{Lev}}(\boldsymbol{S}, \boldsymbol{A}(\boldsymbol{S})) \leq k$.

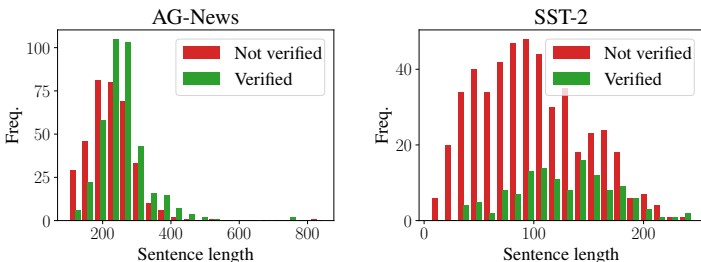

Figure S2: **Sentence length distribution for verified and not verified sentences:** We report the histogram of the lengths for verified and not verified sentences at $k = 1$ with `LipsLev` in the models trained with $p = 2$. Shorter sentences are harder to verify in both SST-2 and AG-News with both `LipsLev` and the brute force approach.

Table S4: **Verified accuracy under bounded $d_{\text{lev}}$:** We report the Clean accuracy (Acc.), Adversarial Accuracy (Adv. Acc.) with Charmer (Abad Rocamora et al., 2024), Verified accuracy (Ver.) and the average runtime in seconds (Time) for the brute-force approach (BruteF), IBP (Huang et al., 2019) and `LipsLev`. **OOT** means the experiment was Out Of Time. ✗ means the method does not support $d_{\text{lev}} > 1$. Our method, `LipsLev`, is the only method able to provide non-trivial verified accuracies for any $k$ in a single forward pass.

| Dataset | $p$ | $k$ | Acc.(%) | Charmer Adv. Acc.(%) | Charmer Time(s) | BruteF Ver.(%) | BruteF Time(s) | IBP Ver.(%) | IBP Time(s) | LipsLev Ver.(%) | LipsLev Time(s) |
|---|---|---|---|---|---|---|---|---|---|---|---|
| AG-News | $\infty$ | 1 | $65.23_{\pm(0.12)}$ | $47.90_{\pm(0.08)}$ | $5.70_{\pm(0.03)}$ | $47.87_{\pm(0.09)}$ | $16.15_{\pm(0.23)}$ | $27.77_{\pm(0.12)}$ | $16.76_{\pm(0.26)}$ | $32.33_{\pm(0.31)}$ | $0.0015_{\pm(0.00033)}$ |
| | | 2 | | $32.97_{\pm(0.38)}$ | $5.70_{\pm(0.03)}$ | | OOT | ✗ | | $11.60_{\pm(0.45)}$ | $0.0015_{\pm(0.00033)}$ |
| | 1 | 1 | $69.63_{\pm(0.19)}$ | $54.47_{\pm(0.49)}$ | $5.43_{\pm(0.33)}$ | $54.43_{\pm(0.53)}$ | $15.33_{\pm(0.34)}$ | $18.93_{\pm(0.50)}$ | $17.56_{\pm(1.62)}$ | $34.50_{\pm(0.36)}$ | $0.00140_{\pm(0.00007)}$ |
| | | 2 | | $37.77_{\pm(0.46)}$ | $5.43_{\pm(0.33)}$ | | OOT | ✗ | | $12.53_{\pm(0.29)}$ | $0.00140_{\pm(0.00007)}$ |
| | 2 | 1 | $74.80_{\pm(0.45)}$ | $62.20_{\pm(0.75)}$ | $7.32_{\pm(0.54)}$ | $62.07_{\pm(0.82)}$ | $29.12_{\pm(1.88)}$ | $29.10_{\pm(0.45)}$ | $31.54_{\pm(0.55)}$ | $38.80_{\pm(0.29)}$ | $0.00970_{\pm(0.00044)}$ |
| | | 2 | | $46.47_{\pm(0.29)}$ | $7.32_{\pm(0.54)}$ | | OOT | ✗ | | $13.93_{\pm(0.21)}$ | $0.00970_{\pm(0.00044)}$ |
| SST-2 | $\infty$ | 1 | $63.95_{\pm(0.30)}$ | $39.68_{\pm(0.99)}$ | $1.84_{\pm(0.05)}$ | $39.68_{\pm(0.99)}$ | $2.27_{\pm(0.079)}$ | $33.94_{\pm(1.11)}$ | $2.88_{\pm(0.092)}$ | $14.68_{\pm(0.25)}$ | $0.00084_{\pm(0.00024)}$ |
| | | 2 | | $19.92_{\pm(1.16)}$ | $1.84_{\pm(0.05)}$ | | OOT | ✗ | | $0.99_{\pm(0.05)}$ | $0.00084_{\pm(0.00024)}$ |
| | 1 | 1 | $69.69_{\pm(0.14)}$ | $45.26_{\pm(0.20)}$ | $1.91_{\pm(0.03)}$ | $45.22_{\pm(0.14)}$ | $2.31_{\pm(0.16)}$ | $19.00_{\pm(1.08)}$ | $2.99_{\pm(0.14)}$ | $18.69_{\pm(0.80)}$ | $0.0022_{\pm(0.0017)}$ |
| | | 2 | | $26.11_{\pm(0.55)}$ | $1.91_{\pm(0.03)}$ | | OOT | ✗ | | $1.83_{\pm(0.00)}$ | $0.0022_{\pm(0.0017)}$ |
| | 2 | 1 | $69.95_{\pm(0.32)}$ | $48.81_{\pm(0.42)}$ | $2.09_{\pm(0.07)}$ | $48.78_{\pm(0.43)}$ | $4.23_{\pm(0.11)}$ | $16.06_{\pm(1.17)}$ | $5.22_{\pm(0.49)}$ | $14.57_{\pm(0.34)}$ | $0.0047_{\pm(0.0023)}$ |
| | | 2 | | $30.70_{\pm(0.81)}$ | $2.09_{\pm(0.07)}$ | | OOT | ✗ | | $0.73_{\pm(0.27)}$ | $0.0047_{\pm(0.0023)}$ |
| Fake-News | $\infty$ | 1 | $100.00_{\pm(0.00)}$ | $86.67_{\pm(0.94)}$ | $66.82_{\pm(1.98)}$ | $86.67_{\pm(0.94)}$ | $972.46_{\pm(8.15)}$ | $85.33_{\pm(0.94)}$ | $989.84_{\pm(8.40)}$ | $85.33_{\pm(0.94)}$ | $0.017_{\pm(0.0067)}$ |
| | | 2 | | $76.00_{\pm(1.63)}$ | $66.82_{\pm(1.98)}$ | | OOT | ✗ | | $68.67_{\pm(0.94)}$ | $0.017_{\pm(0.0067)}$ |
| | 1 | 1 | $98.00_{\pm(1.63)}$ | $92.00_{\pm(0.00)}$ | $67.11_{\pm(1.87)}$ | $92.00_{\pm(0.00)}$ | $978.94_{\pm(15.91)}$ | $91.33_{\pm(0.94)}$ | $990.32_{\pm(14.85)}$ | $91.33_{\pm(0.94)}$ | $0.014_{\pm(0.0056)}$ |
| | | 2 | | $79.33_{\pm(2.49)}$ | $67.11_{\pm(1.87)}$ | | OOT | ✗ | | $75.33_{\pm(3.40)}$ | $0.014_{\pm(0.0056)}$ |
| | 2 | 1 | $98.00_{\pm(1.63)}$ | $88.67_{\pm(4.99)}$ | $73.52_{\pm(2.77)}$ | $88.67_{\pm(4.99)}$ | $1224.45_{\pm(8.66)}$ | $87.33_{\pm(4.11)}$ | $1466.38_{\pm(294.21)}$ | $87.33_{\pm(6.80)}$ | $0.0089_{\pm(0.010)}$ |
| | | 2 | | $78.00_{\pm(4.32)}$ | $73.52_{\pm(2.77)}$ | | OOT | ✗ | | $71.33_{\pm(5.25)}$ | $0.0089_{\pm(0.010)}$ |
| IMDB | $\infty$ | 1 | $74.57_{\pm(5.22)}$ | $67.43_{\pm(4.70)}$ | $14.16_{\pm(0.40)}$ | $67.43_{\pm(4.70)}$ | $130.49_{\pm(3.38)}$ | $61.50_{\pm(4.73)}$ | $138.20_{\pm(6.12)}$ | $31.37_{\pm(4.54)}$ | $0.0047_{\pm(0.0015)}$ |
| | | 2 | | $59.77_{\pm(4.81)}$ | $14.16_{\pm(0.40)}$ | | OOT | ✗ | | $5.90_{\pm(1.36)}$ | $0.0047_{\pm(0.0015)}$ |
| | 1 | 1 | $69.57_{\pm(7.18)}$ | $61.17_{\pm(8.92)}$ | $14.44_{\pm(0.27)}$ | $61.00_{\pm(8.82)}$ | $134.23_{\pm(1.64)}$ | $47.30_{\pm(10.51)}$ | $135.22_{\pm(0.31)}$ | $28.73_{\pm(6.94)}$ | $0.0027_{\pm(0.0025)}$ |
| | | 2 | | $52.20_{\pm(9.33)}$ | $14.44_{\pm(0.27)}$ | | OOT | ✗ | | $6.80_{\pm(2.16)}$ | $0.0027_{\pm(0.0025)}$ |
| | 2 | 1 | $60.60_{\pm(4.21)}$ | $46.87_{\pm(0.62)}$ | $16.24_{\pm(0.49)}$ | $46.73_{\pm(0.78)}$ | $261.99_{\pm(62.11)}$ | $37.57_{\pm(6.35)}$ | $308.73_{\pm(1.70)}$ | $8.67_{\pm(5.08)}$ | $0.0019_{\pm(0.0011)}$ |
| | | 2 | | $35.10_{\pm(3.36)}$ | $16.24_{\pm(0.49)}$ | | OOT | ✗ | | $0.87_{\pm(0.66)}$ | $0.0019_{\pm(0.0011)}$ |

Table S5: **`LipsLev` verified accuracy in FakeNews over the first $1,000$ validation samples and up to $k = 10$.** We observe our method is able to verify non-trivial accuracy with even up to 10 character changes.

| $p$ | Clean Acc. | 1 | 2 | 3 | 4 | Verified Acc. 5 | 6 | 7 | 8 | 9 | 10 |
|---|---|---|---|---|---|---|---|---|---|---|---|
| $\infty$ | $95.63_{\pm(0.19)}$ | $87.50_{\pm(0.42)}$ | $72.43_{\pm(0.31)}$ | $49.37_{\pm(1.54)}$ | $31.03_{\pm(1.33)}$ | $20.03_{\pm(1.21)}$ | $14.07_{\pm(1.33)}$ | $9.77_{\pm(1.11)}$ | $6.67_{\pm(0.83)}$ | $5.00_{\pm(0.85)}$ | $3.60_{\pm(0.57)}$ |
| 1 | $95.90_{\pm(0.16)}$ | $88.93_{\pm(1.72)}$ | $75.97_{\pm(2.91)}$ | $56.33_{\pm(4.24)}$ | $38.50_{\pm(4.82)}$ | $24.13_{\pm(4.14)}$ | $15.83_{\pm(3.79)}$ | $11.30_{\pm(3.40)}$ | $7.47_{\pm(2.40)}$ | $5.50_{\pm(2.01)}$ | $4.27_{\pm(1.54)}$ |
| 2 | $95.00_{\pm(0.92)}$ | $87.50_{\pm(2.70)}$ | $70.77_{\pm(7.90)}$ | $48.37_{\pm(11.71)}$ | $30.97_{\pm(10.06)}$ | $20.57_{\pm(6.44)}$ | $13.07_{\pm(4.89)}$ | $9.20_{\pm(4.27)}$ | $6.10_{\pm(2.73)}$ | $4.57_{\pm(2.02)}$ | $3.50_{\pm(1.71)}$ |

## A.5 HYPERPARAMETER SELECTION

In order to select the best learning rate in each dataset and $p$ norm for the ERP distance, we compute the clean and verified accuracy at $k = 1$ in a validation set of $1,000$ samples extracted from each training set. We test the learning rate values $\{0.1, 0.5, 1, 5, 10, 50, 100, 500, 1000\}$. We train convolutional models with 1 convolutional layer and the standard embedding, hidden and kernel sizes in Section 5.1. We notice these large learning rates are needed due to the 1-Lipschitz formulation in Eq. (7).

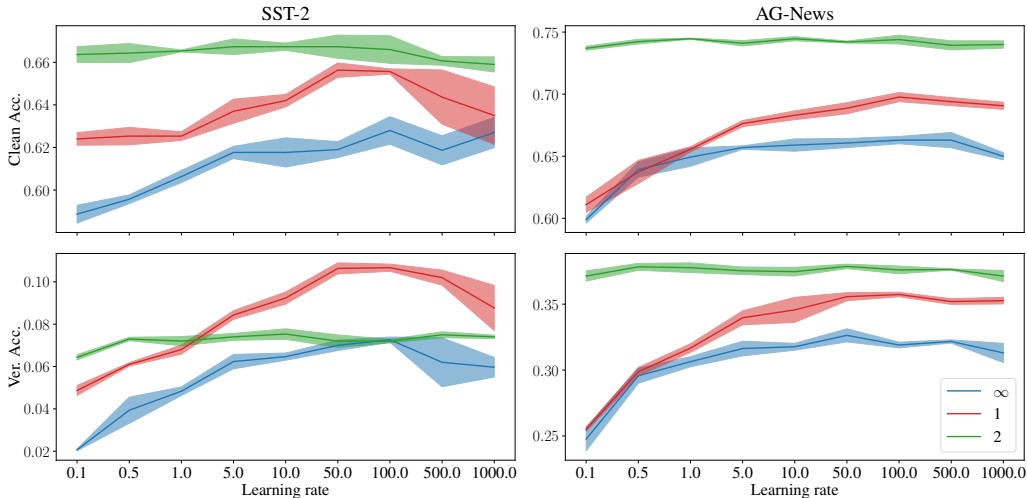

Figure S3: **Learning rate selection for the SST-2 and AG-News datasets:** We report the clean and verified accuracy in a validation set of 1,000 sentences extracted from the training split of each dataset and set aside during training. We set the learning rate equal to 100 in the rest of our experiments as it provides a good trade-off between clean and verified accuracy for all norms and datasets.

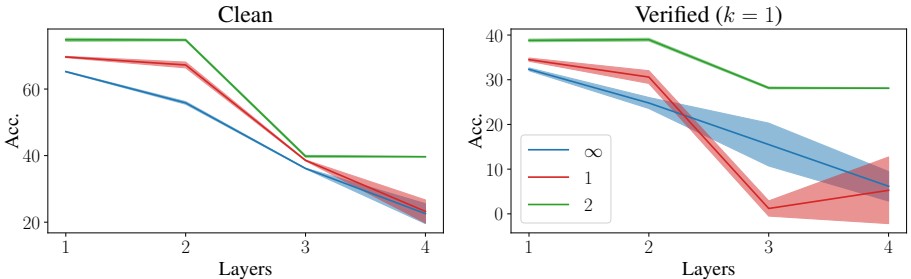

Figure S4: **Training deeper models in AG-News:** We report the clean and verified accuracies with `LipsLev` at $k = 1$ for $p \in \{1, 2, \infty\}$. Clean and verified accuracies decrease with the number of layers. With $p = 2$ the performance is less degraded with the number of layers.

Based on the results from Fig. S3, we select 100 as our learning rate for the rest of experiments in this work.

## A.6 TRAINING DEEPER MODELS

In this section, we study the performance of models with more than one convolutional layer. We train with $1, 2, 3$ and $4$ convolutional layers with a hidden size of $100$ and a kernel size of $5$ and $10$ for SST-2 and AG-News respectively. We train the models with the 1-Lipschitz formulation in Eq. (7) with $p \in \{1, 2, \infty\}$.

In Figs. S4 and S5 we can observe that increasing the number of layers degrades the clean and verified accuracy for every value of $p$. Nevertheless, for $p = 2$, the effect is diminished. Jointly with the improved performance when using $p = 2$ in Section 5.2, we advocate for its use in the ERP distance. We believe this performance degradation is related to the gradient attenuation phenomenon (Li et al., 2019). It remains an open problem to avoid gradient attenuation in the case where the Lipschitz constant of the ERP distance is enforced to be 1.

In Table S6 we can observe that our models have low latencies. Noticeably, with $p = 2$ we observe a larger latency than with $p \in \{1, \infty\}$. This is due to the need to compute the espectral norm at each

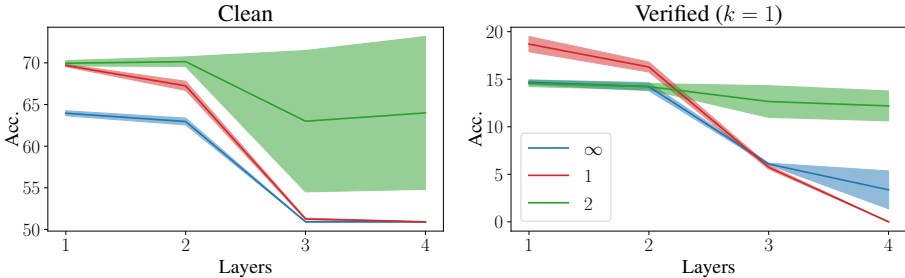

Figure S5: **Training deeper models in SST-2:** We report the clean and verified accuracies with `LipsLev` at $k = 1$ for $p \in \{1, 2, \infty\}$. Clean and verified accuracies decrease with the number of layers. With $p = 2$ the performance is less degraded with the number of layers.

iteration. Nevertheless, this cost is still low and only incurred during training, as by rescaling the weights, we can any model in Eq. (7) formulation as a model in Eq. (6).

Table S6: **Latency in seconds for different $p$ and number of layers.**

| $p$ | Layers | | | |
| --- | --- | --- | --- | --- |
| | 1 | 2 | 3 | 4 |
| 1 | $0.0017_{\pm(0.0179)}$ | $0.0015_{\pm(0.0007)}$ | $0.0015_{\pm(0.0003)}$ | $0.0019_{\pm(0.0002)}$ |
| 2 | $0.0225_{\pm(0.0045)}$ | $0.0393_{\pm(0.0026)}$ | $0.0567_{\pm(0.0028)}$ | $0.0740_{\pm(0.0036)}$ |
| $\infty$ | $0.0007_{\pm(0.0000)}$ | $0.0011_{\pm(0.0000)}$ | $0.0015_{\pm(0.0000)}$ | $0.0019_{\pm(0.0000)}$ |

### A.7 COMPARISON AGAINST RANDOMIZED SMOOTHING

In this section, we analyze the differences in methodology and results, between our approach and the Randomized Smoothing approach in Huang et al. (2023): RS-Del.

As an advantage, RS-Del can be employed with any base classifier $\boldsymbol{f} : \mathcal{S}(\Gamma) \rightarrow \mathbb{R}^o$ assigning scores to each of the $o$ classes, without restrictions on the architecture. Nevertheless, neither the classifier verified by RS-Del, nor the certified radius itself are deterministic. In Randomized Smoothing, given a classifier $\boldsymbol{f}$ and given a set of perturbations $\mathcal{P}(\boldsymbol{S})$, a smoothed classifier $\tilde{\boldsymbol{f}}(\boldsymbol{S}) = \underset{\tilde{\boldsymbol{S}} \sim \text{Unif.}(\mathcal{P}(\boldsymbol{S}))}{\mathbb{E}} \left[ \boldsymbol{f}(\tilde{\boldsymbol{S}}) \right]$ is constructed.

In practice, multiple perturbations ($n_{\text{pred}}$) are sampled from $\mathcal{P}(\boldsymbol{S})$ and the prediction is estimated through Monte-Carlo as the average $\tilde{\boldsymbol{f}}(\boldsymbol{S}) \approx \sum_{i=1}^{n_{\text{pred}}} \boldsymbol{f}(\tilde{\boldsymbol{S}}_i)$, with $\tilde{\boldsymbol{S}}_i$ sampled i.i.d uniformly from $\mathcal{P}(\boldsymbol{S})$. Similarly, an $\alpha$-confidence lower bound of the margin is estimated with $n_{\text{bnd}}$ samples, which is then used to obtain an $\alpha$-confidence estimate of the certified radius. Therefore, the certificates provided with Randomized Smoothing are probabilistic and regarding the smoothed classifier $\tilde{\boldsymbol{f}}$, not the original $\boldsymbol{f}$.

Contrarily, `LipsLev` imposes the strong constraint of knowing an upper bound of the Lipschitz constant of the margins of $\boldsymbol{f}$. However, this enables the direct, non-probabilistic, estimate of the certified radius in a single forward pass through $\boldsymbol{f}$.

Having these differences in mind, we compare the performance of `LipsLev` and RD-Del when verifying the models in Table 2. To do so, we run RS-Del with the recommended $n_{\text{pred}} = 1,000$, $n_{\text{bnd}} = 4,000$, $p_{\text{del}} = 0.9$ and $\alpha = 0.05$.

In Table S7 we can observe that RS-Del significantly degrades the clean accuracy of the classifier, this is due to the randomization introduced in the prediction process. However, RS-Del is able to improve the verified accuracy in SST-2 for all $p$ and in IMDB for $p = 2$. In terms of runtime, we observe that, while being significantly slower than `LipsLev`, RS-Del is able to provide certificates in a time range from 1 to 2 seconds, which is considerably faster than Brute-F and IBP.

Table S7: **Comparison against Randomized Smoothing:** We report the Clean accuracy (Acc.), Verified accuracy (Ver.) and the average runtime in seconds (Time) for `LipsLev` and RS-Del (Huang et al., 2023). RS-Del reduces drops significantly the clean accuracy of the classifier. RS-Del improves the verified accuracy of `LipsLev` in SST-2 for all $p$ and IMDB for $p = 2$.

| Dataset | $p$ | $k$ | LipsLev Acc.(%) | LipsLev Ver.(%) | LipsLev Time(s) | RS-Del (Huang et al., 2023) Acc.(%) | RS-Del (Huang et al., 2023) Ver.(%) | RS-Del (Huang et al., 2023) Time(s) |
|---|---|---|---|---|---|---|---|---|
| AG-News | $\infty$ | 1 | $65.23_{\pm(0.12)}$ | $32.33_{\pm(0.31)}$ | $0.0015_{\pm(0.00033)}$ | $51.77_{\pm(0.17)}$ | $2.57_{\pm(0.41)}$ | $1.08_{\pm(0.06)}$ |
| | | 2 | | $11.60_{\pm(0.45)}$ | $0.0015_{\pm(0.00033)}$ | | $0.27_{\pm(0.05)}$ | $1.08_{\pm(0.06)}$ |
| | 1 | 1 | $69.63_{\pm(0.19)}$ | $34.50_{\pm(0.36)}$ | $0.00140_{\pm(0.00007)}$ | $50.67_{\pm(0.90)}$ | $3.73_{\pm(0.19)}$ | $1.05_{\pm(0.05)}$ |
| | | 2 | | $12.53_{\pm(0.29)}$ | $0.00140_{\pm(0.00007)}$ | | $0.80_{\pm(0.22)}$ | $1.05_{\pm(0.05)}$ |
| | 2 | 1 | $74.80_{\pm(0.45)}$ | $38.80_{\pm(0.29)}$ | $0.00970_{\pm(0.00044)}$ | $48.20_{\pm(0.37)}$ | $4.60_{\pm(0.41)}$ | $1.68_{\pm(0.07)}$ |
| | | 2 | | $13.93_{\pm(0.21)}$ | $0.00970_{\pm(0.00044)}$ | | $0.90_{\pm(0.16)}$ | $1.68_{\pm(0.07)}$ |
| SST-2 | $\infty$ | 1 | $63.95_{\pm(0.30)}$ | $14.68_{\pm(0.25)}$ | $0.00084_{\pm(0.00024)}$ | $57.11_{\pm(0.49)}$ | $21.98_{\pm(1.37)}$ | $0.60_{\pm(0.04)}$ |
| | | 2 | | $0.99_{\pm(0.05)}$ | $0.00084_{\pm(0.00024)}$ | | $3.78_{\pm(1.12)}$ | $0.60_{\pm(0.04)}$ |
| | 1 | 1 | $69.69_{\pm(0.14)}$ | $18.69_{\pm(0.80)}$ | $0.0022_{\pm(0.0017)}$ | $56.42_{\pm(1.08)}$ | $21.10_{\pm(1.95)}$ | $0.69_{\pm(0.13)}$ |
| | | 2 | | $1.83_{\pm(0.00)}$ | $0.0022_{\pm(0.0017)}$ | | $3.82_{\pm(0.76)}$ | $0.69_{\pm(0.13)}$ |
| | 2 | 1 | $69.95_{\pm(0.32)}$ | $14.57_{\pm(0.34)}$ | $0.0047_{\pm(0.0023)}$ | $59.17_{\pm(0.99)}$ | $22.40_{\pm(0.98)}$ | $0.94_{\pm(0.06)}$ |
| | | 2 | | $0.73_{\pm(0.27)}$ | $0.0047_{\pm(0.0023)}$ | | $4.86_{\pm(0.42)}$ | $0.94_{\pm(0.06)}$ |
| Fake-News | $\infty$ | 1 | $100.00_{\pm(0.00)}$ | $85.33_{\pm(0.94)}$ | $0.017_{\pm(0.0067)}$ | $80.67_{\pm(0.94)}$ | $69.33_{\pm(2.49)}$ | $1.87_{\pm(0.10)}$ |
| | | 2 | | $68.67_{\pm(0.94)}$ | $0.017_{\pm(0.0067)}$ | | $56.67_{\pm(0.94)}$ | $1.87_{\pm(0.10)}$ |
| | 1 | 1 | $98.00_{\pm(1.63)}$ | $91.33_{\pm(0.94)}$ | $0.014_{\pm(0.0056)}$ | $77.33_{\pm(1.89)}$ | $66.00_{\pm(2.83)}$ | $1.87_{\pm(0.06)}$ |
| | | 2 | | $75.33_{\pm(3.40)}$ | $0.014_{\pm(0.0056)}$ | | $54.00_{\pm(1.63)}$ | $1.87_{\pm(0.06)}$ |
| | 2 | 1 | $98.00_{\pm(1.63)}$ | $87.33_{\pm(6.80)}$ | $0.0089_{\pm(0.010)}$ | $75.33_{\pm(8.22)}$ | $62.67_{\pm(8.99)}$ | $2.24_{\pm(0.11)}$ |
| | | 2 | | $71.33_{\pm(5.25)}$ | $0.0089_{\pm(0.010)}$ | | $57.33_{\pm(6.60)}$ | $2.24_{\pm(0.11)}$ |
| IMDB | $\infty$ | 1 | $74.57_{\pm(5.22)}$ | $31.37_{\pm(4.54)}$ | $0.0047_{\pm(0.0015)}$ | $6.90_{\pm(0.00)}$ | $0.90_{\pm(0.00)}$ | $1.64_{\pm(0.00)}$ |
| | | 2 | | $5.90_{\pm(1.36)}$ | $0.0047_{\pm(0.0015)}$ | | $0.00_{\pm(0.00)}$ | $1.64_{\pm(0.00)}$ |
| | 1 | 1 | $69.57_{\pm(7.18)}$ | $28.73_{\pm(6.94)}$ | $0.0027_{\pm(0.0025)}$ | $19.20_{\pm(0.00)}$ | $3.50_{\pm(0.00)}$ | $1.74_{\pm(0.00)}$ |
| | | 2 | | $6.80_{\pm(2.16)}$ | $0.0027_{\pm(0.0025)}$ | | $0.40_{\pm(0.00)}$ | $1.74_{\pm(0.00)}$ |
| | 2 | 1 | $60.60_{\pm(4.21)}$ | $8.67_{\pm(5.08)}$ | $0.0019_{\pm(0.0011)}$ | $48.36_{\pm(36.43)}$ | $33.67_{\pm(27.15)}$ | $2.14_{\pm(0.20)}$ |
| | | 2 | | $0.87_{\pm(0.66)}$ | $0.0019_{\pm(0.0011)}$ | | $18.43_{\pm(15.85)}$ | $2.14_{\pm(0.20)}$ |

## B  PROOFS

In this section we introduce the mathematical tools needed to derive our Lipschitz constant upper bounds for each layer in Eq. (6). The section concludes with the proof of our main result in Theorem 4.3. In Appendix B.1 we present some remarks to be considered regarding global and local Lipschitz constants.

**Definition S1** (Zero-paddings). Let $\boldsymbol{X} \in \mathcal{X}_d$ a sequence of $m$ non-zero vectors. Let $l \geq m$, a zero padding function $\boldsymbol{Z} : \mathcal{X}_d \to \mathbb{R}^{l \times d}$ is some function defined by the tuple:

$$(i_k)_{k=1}^l : \begin{cases} m \geq i_k > i_j \ \forall 1 < j < k & \text{if } i_k \neq 0 \\ |\{k \in [l] : i_k = 0\}| = l - m & \text{if } i_k = 0 \end{cases}$$

so that:

$$\boldsymbol{z}_k(\boldsymbol{X}) = \begin{cases} \boldsymbol{x}_{i_k} & \text{if } i_k \neq 0 \\ \boldsymbol{0} & \text{if } i_k = 0 \end{cases}$$

Intuitively, a valid zero-padding function inserts $l - m$ zeros in between any vector of the sequence, the beginning or the end. We denote as $\mathcal{Z}_{m,l}$ the set of zero paddings from sequences of length $m$ to sequences of length $l$.

*Remark* S2. Given a matrix $\boldsymbol{A} \in \mathbb{R}^{m \times m}$ and a zero padding $\boldsymbol{Z} \in \mathcal{Z}_{m,l}$, we denote the column and row-wise padding as $\overline{\boldsymbol{Z}}(\boldsymbol{A}) = \boldsymbol{Z}(\boldsymbol{Z}(\boldsymbol{A}^\top)^\top) \in \mathbb{R}^{l \times l}$.

**Proposition S3** (Alternative definition of $d_{\text{ERP}}^p$). *Let $d_{\text{ERP}}^p$ be as in Definition 4.1. Let $\boldsymbol{A} \in \mathbb{R}^{m \times d}$ and $\boldsymbol{B} \in \mathbb{R}^{n \times d}$ be two sequences. Let $\mathcal{Z}_{m,m+n}$ and $\mathcal{Z}_{n,m+n}$ be the zero-padding functions from length $m$ and $n$ respectively to length $m + n$. The ERP distance can be expressed as:*

$$d_{ERP}^p(\boldsymbol{A}, \boldsymbol{B}) = \min_{\boldsymbol{Z}^a \in \mathcal{Z}_{m,m+n}, \boldsymbol{Z}^b \in \mathcal{Z}_{n,m+n}} \sum_{k=1}^{m+n} \left\| \boldsymbol{z}_k^a(\boldsymbol{A}) - \boldsymbol{z}_k^b(\boldsymbol{B}) \right\|_p$$

**Lemma S4** (Properties of the ERP distance). *Some important properties of the ERP distance are summarized here:*

*(a) Generalization of edit distance:*
*In the case of having sequences of one-hot vectors $\boldsymbol{A} \in \{0,1\}^{m \times d} : ||\boldsymbol{a}_i||_1 = 1$, and using $p = \infty$, the ERP distance is equal to the edit distance (Levenshtein et al., 1966).*

*(b) Invariance to the concatenation of zeros:*

$$d_{ERP}^p(\boldsymbol{A} \oplus \boldsymbol{0}, \boldsymbol{B}) = d_{ERP}^p(\boldsymbol{0} \oplus \boldsymbol{A}, \boldsymbol{B}) = d_{ERP}^p(\boldsymbol{A}, \boldsymbol{B}) \;\; \forall \boldsymbol{A} \in \mathbb{R}^{m \times d}, \boldsymbol{B} \in \mathbb{R}^{n \times d}$$

*(c) Distance to the empty set:*

$$d_{ERP}^p(\boldsymbol{A}, \emptyset) = \sum_{i=1}^{m} ||\boldsymbol{a}_i||_p \;\; \forall \boldsymbol{A} \in \mathbb{R}^{m \times d}$$

*(d) Symmetry:*
$$d_{ERP}^p(\boldsymbol{A}, \boldsymbol{B}) = d_{ERP}^p(\boldsymbol{B}, \boldsymbol{A}) \;\; \forall \boldsymbol{A} \in \mathbb{R}^{m \times d}, \boldsymbol{B} \in \mathbb{R}^{n \times d}$$

*(e) Triangular inequality:*
*For any $\boldsymbol{A} \in \mathbb{R}^{m \times d}, \boldsymbol{B} \in \mathbb{R}^{n \times d}, \boldsymbol{B} \in \mathbb{R}^{l \times d}$, we have:*
$$d_{ERP}^p(\boldsymbol{A}, \boldsymbol{B}) \leq d_{ERP}^p(\boldsymbol{A}, \boldsymbol{C}) + d_{ERP}^p(\boldsymbol{C}, \boldsymbol{B}).$$

*(f) Subdistance:*
*The ERP distance is not a distance because of its invariance to the concatenation of zeros:*
$$d_{ERP}^p(\boldsymbol{A}, \boldsymbol{A} \oplus \boldsymbol{0}) = d_{ERP}^p(\boldsymbol{A}, \boldsymbol{A}) = 0 \;\; \forall \boldsymbol{A} \in \mathbb{R}^{m \times d}$$

*proof of Lemma S4.* Properties (a), (b), (c), (d) and (f) are straightforward from the deffinition, we will prove the triangular inequality (e). This proof follows similarly to the one of Waterman et al. (1976) for the standard Levenshtein distance. Let $L = m + n + l$, starting from the definition in Proposition S3:

$$d_{\text{ERP}}^p(\boldsymbol{A}, \boldsymbol{B}) + d_{\text{ERP}}^p(\boldsymbol{B}, \boldsymbol{C}) = \min_{\substack{\boldsymbol{Z}^a \in \mathcal{Z}_{m,L}, \boldsymbol{Z}^b \in \mathcal{Z}_{n,L} \\ \boldsymbol{Z}^c \in \mathcal{Z}_{n,L}, \boldsymbol{Z}^d \in \mathcal{Z}_{l,L}}} \sum_{k=1}^{L} \left| \left| \boldsymbol{z}_k^a(\boldsymbol{A}) - \boldsymbol{z}_k^b(\boldsymbol{B}) \right| \right|_p$$
$$+ \sum_{j=1}^{L} \left| \left| \boldsymbol{z}_j^c(\boldsymbol{B}) - \boldsymbol{z}_j^d(\boldsymbol{C}) \right| \right|_p .$$

Let $\boldsymbol{Z}^e, \boldsymbol{Z}^f \in \mathcal{Z}_{L,2L}$ be two zero paddings so that $\boldsymbol{Z}^e(\boldsymbol{Z}^b(B)) = \boldsymbol{Z}^f(\boldsymbol{Z}^c(B))$:

$$d_{\text{ERP}}^p(\boldsymbol{A}, \boldsymbol{B}) + d_{\text{ERP}}^p(\boldsymbol{B}, \boldsymbol{C}) = \min_{\substack{\boldsymbol{Z}^a \in \mathcal{Z}_{m,L}, \boldsymbol{Z}^b \in \mathcal{Z}_{n,L} \\ \boldsymbol{Z}^c \in \mathcal{Z}_{n,L}, \boldsymbol{Z}^d \in \mathcal{Z}_{l,L}}} \sum_{k=1}^{2L} \left| \left| \boldsymbol{z}_k^e(\boldsymbol{Z}^a(\boldsymbol{A})) - \boldsymbol{z}_k^e(\boldsymbol{Z}^b(\boldsymbol{B})) \right| \right|_p$$
$$+ \left| \left| \boldsymbol{z}_k^f(\boldsymbol{Z}^c(\boldsymbol{B})) - \boldsymbol{z}_k^f(\boldsymbol{Z}^d(\boldsymbol{C})) \right| \right|_p$$

$$\text{[Triangular ineq. for } ||\cdot||_p] \geq \min_{\substack{\boldsymbol{Z}^a \in \mathcal{Z}_{m,L}, \boldsymbol{Z}^b \in \mathcal{Z}_{n,L} \\ \boldsymbol{Z}^c \in \mathcal{Z}_{n,L}, \boldsymbol{Z}^d \in \mathcal{Z}_{l,L}}} \sum_{k=1}^{2L} \left| \left| \boldsymbol{z}_k^e(\boldsymbol{Z}^a(\boldsymbol{A})) - \boldsymbol{z}_k^e(\boldsymbol{Z}^b(\boldsymbol{B})) \right. \right.$$
$$\left. \left. + \; \boldsymbol{z}_k^f(\boldsymbol{Z}^c(\boldsymbol{B})) - \boldsymbol{z}_k^f(\boldsymbol{Z}^d(\boldsymbol{C})) \right| \right|_p$$

$$[\boldsymbol{Z}^e(\boldsymbol{Z}^b(B)) = \boldsymbol{Z}^f(\boldsymbol{Z}^c(B))] = \min_{\boldsymbol{Z}^a \in \mathcal{Z}_{m,L}, \boldsymbol{Z}^d \in \mathcal{Z}_{l,L}} \sum_{k=1}^{2L} \left| \left| \boldsymbol{z}_k^e(\boldsymbol{Z}^a(\boldsymbol{A})) - \boldsymbol{z}_k^f(\boldsymbol{Z}^d(\boldsymbol{C})) \right| \right|_p$$
$$= d_{\text{ERP}}^p(\boldsymbol{A}, \boldsymbol{C}) \,,$$

where the last equality follows from $\boldsymbol{z}_k^e(\boldsymbol{Z}^a)$ and $\boldsymbol{z}_k^f(\boldsymbol{Z}^d)$ being valid zero paddings. □

**Lemma S5** (Difference of sums). *Let $A, B \in \mathcal{X}_d$, we have that:*

$$\left\| \sum_{i=1}^{m} a_i - \sum_{j=1}^{n} b_j \right\|_p \le d_{ERP}^p(A, B)$$

*proof of Lemma S5.*

$$\left\| \sum_{i=1}^{m} a_i - \sum_{j=1}^{n} b_j \right\|_p = \left\| \min_{Z^a \in \mathcal{Z}_{m,m+n}, Z^b \in \mathcal{Z}_{n,m+n}} \sum_{k=1}^{m+n} z_k^a(A) - z_k^b(B) \right\|_p$$

$$\le \min_{Z^a \in \mathcal{Z}_{m,m+n}, Z^b \in \mathcal{Z}_{n,m+n}} \sum_{k=1}^{m+n} \left\| z_k^a(A) - z_k^b(B) \right\|_p$$

$$= d_{ERP}^p(A, B)$$

$\square$

**Lemma S6** (Difference of means). *Let $A, B \in \mathcal{X}_d$, we have that:*

$$\left\| \frac{1}{m} \cdot \sum_{i=1}^{m} a_i - \frac{1}{n} \cdot \sum_{j=1}^{n} b_j \right\|_p \le \frac{|m-n|}{m \cdot n} \cdot \left\| \sum_{i=1}^{m} a_i \right\|_p + \frac{1}{n} \cdot d_{ERP}^p(A, B)$$

*and*

$$\left\| \frac{1}{m} \cdot \sum_{i=1}^{m} a_i - \frac{1}{n} \cdot \sum_{j=1}^{n} b_j \right\|_p \le \frac{|m-n|}{m \cdot n} \cdot \left\| \sum_{j=1}^{m} b_j \right\|_p + \frac{1}{m} \cdot d_{ERP}^p(A, B) .$$

*In the case of $A$ and $B$ being sequences of one-hot vectors, we have that:*

$$\left\| \frac{1}{m} \cdot \sum_{i=1}^{m} a_i - \frac{1}{n} \cdot \sum_{j=1}^{n} b_j \right\|_\infty \le \begin{cases} \frac{1}{m} \cdot d_{lev}(A, B) & \text{if } m = n \\ \frac{2}{m} \cdot d_{lev}(A, B) & \text{if } m \ne n \end{cases}$$

*proof of Lemma S6.* Starting with the first result:

$$\left\| \frac{1}{m} \cdot \sum_{i=1}^{m} a_i - \frac{1}{n} \cdot \sum_{j=1}^{n} b_j \right\|_p = \frac{1}{m \cdot n} \left\| (n + m - m) \cdot \sum_{i=1}^{m} a_i - m \cdot \sum_{j=1}^{n} b_j \right\|_p$$

$$\le \frac{1}{m \cdot n} \left( |m-n| \cdot \left\| \sum_{i=1}^{m} a_i \right\|_p + m \cdot \left\| \sum_{i=1}^{m} a_i - \sum_{j=1}^{n} b_j \right\|_p \right)$$

$$[\text{Lemma S5}] \le \frac{|m-n|}{m \cdot n} \cdot \left\| \sum_{i=1}^{m} a_i \right\|_p + \frac{1}{n} \cdot d_{ERP}^p(A, B) .$$

Note that since $A$ and $B$ are interchangeable, we immediately have:

$$\left\| \frac{1}{m} \cdot \sum_{i=1}^{m} a_i - \frac{1}{n} \cdot \sum_{j=1}^{n} b_j \right\|_p \le \frac{|m-n|}{m \cdot n} \cdot \left\| \sum_{j=1}^{n} b_j \right\|_p + \frac{1}{m} \cdot d_{ERP}^p(A, B) . \tag{8}$$

In the case $A$ and $B$ are sequences of one-hot vectors, if $m = n$, we can directly get the $1/m$ factor out of the norm and apply Lemma S5 to get the first case. For the case $m \ne n$, we can manipulate

Eq. (8) to get the desired result:

$$\left\|\frac{1}{m}\cdot\sum_{i=1}^{m}\boldsymbol{a}_i - \frac{1}{n}\cdot\sum_{j=1}^{n}\boldsymbol{b}_j\right\|_{\infty} \leq \frac{|m-n|}{m\cdot n}\cdot\left\|\sum_{j=1}^{n}\boldsymbol{b}_j\right\|_{\infty} + \frac{1}{m}\cdot d_{\text{lev}}\left(\boldsymbol{A},\boldsymbol{B}\right)$$

$$[|m-n| \leq d_{\text{lev}}\left(\boldsymbol{A},\boldsymbol{B}\right) + \text{Triang. ineq.}] \leq \left(\frac{1}{m\cdot n}\cdot\sum_{j=1}^{n}||\boldsymbol{b}_j||_{\infty} + \frac{1}{m}\right)\cdot d_{\text{lev}}\left(\boldsymbol{A},\boldsymbol{B}\right)$$

$$[||\boldsymbol{b}_j||_{\infty} = 1 \ \forall j \in [n]] = \frac{2}{m}\cdot d_{\text{lev}}\left(\boldsymbol{A},\boldsymbol{B}\right) \ .$$

$\square$

**Lemma S7** (Linear transformations). *Let $\boldsymbol{A}, \boldsymbol{B} \in \mathcal{X}_d$ be two sequences and $V \in \mathbb{R}^{d \times k}$. We have that:*

$$d_{ERP}^p(\boldsymbol{A}\boldsymbol{V}, \boldsymbol{B}\boldsymbol{V}) \leq d_{ERP}^p(\boldsymbol{A}, \boldsymbol{B})\,||\boldsymbol{V}||_p$$

*In the case of sequences of one-hot vectors, we have that:*

$$d_{ERP}^p(\boldsymbol{A}\boldsymbol{V}, \boldsymbol{B}\boldsymbol{V}) \leq d_{Lev}(\boldsymbol{A}, \boldsymbol{B}) \cdot M(\boldsymbol{V}) \ ,$$

*where*

$$M(\boldsymbol{V}) = \max\{\max_{i \in [d]}||\boldsymbol{v}_i||_p\,, \max_{i,j \in [d]}||\boldsymbol{v}_i - \boldsymbol{v}_j||_p\}$$

*Proof.* Follows immediately from Definition 4.1 and the fact that $||\boldsymbol{A}\boldsymbol{B}|| \leq ||\boldsymbol{A}||\,||\boldsymbol{B}||$ for any matrices $\boldsymbol{A}$ and $\boldsymbol{B}$. The second result for one-hot vectors follows immediately from the fact that the biggest change in the embedding sequence that can be produced from a single-character change, is either given by inserting the character with the largest norm embedding (left side of the max), or replacing a character with the character that has the furthest away embedding in the $\ell_p$ norm (left side of the max). $\square$

**Lemma S8** (Elementwise Lipschitz functions). *Let $d_{ERP}^p$ be as in Definition 4.1. Let $\boldsymbol{A} \in \mathbb{R}^{m \times d}$ and $\boldsymbol{B} \in \mathbb{R}^{n \times d}$ be two sequences. Let $f : \mathbb{R}^d \to \mathbb{R}^k$ be a Lipschitz function so that:*

$$||\boldsymbol{f}(\boldsymbol{a}) - \boldsymbol{f}(\boldsymbol{b})||_p \leq L_f \cdot ||\boldsymbol{a} - \boldsymbol{b}||_p \ \ \forall \boldsymbol{a}, \boldsymbol{b} \in \mathbb{R}^d \ .$$

*Let $\boldsymbol{F}(\boldsymbol{A}) \in \mathbb{R}^{m \times k}$ and $\boldsymbol{F}(\boldsymbol{B}) \in \mathbb{R}^{n \times k}$ be the application of $f$ to every vector in both sequences, we immediately have that:*

$$d_{ERP}^p(\boldsymbol{F}(\boldsymbol{A}), \boldsymbol{F}(\boldsymbol{B})) \leq L_f \cdot d_{ERP}^p(\boldsymbol{A}, \boldsymbol{B})$$

**Lemma S9** (Convolution). *Let $d_{ERP}^p$ be as in Definition 4.1. Let $\boldsymbol{P} \in \{0,1\}^{m \times d}$ and $\boldsymbol{Q} \in \{0,1\}^{n \times d}$ be two sequences of $m$ and $n$ one hot-vectors respectively. Let the function working with arbitrary sequence length $l$ be $\boldsymbol{F} : \{0,1\}^{l \times d} \to \mathbb{R}^{l \times r}$. Let the convolutional filter $\boldsymbol{C} : \mathbb{R}^{l \times r} \to \mathbb{R}^{(l+q-1) \times k}$ with kernel $\mathcal{K} \in \mathbb{R}^{q \times k \times r}$, where $q$ is the kernel size and $k$ is the number of filters. We have that:*

$$d_{ERP}^p\left(\boldsymbol{C}(\boldsymbol{F}(\boldsymbol{P})), \boldsymbol{C}(\boldsymbol{F}(\boldsymbol{Q}))\right) \ \leq M(\mathcal{K}) \cdot d_{ERP}^p\left(\boldsymbol{F}(\boldsymbol{P}), \boldsymbol{F}(\boldsymbol{Q})\right) \ .$$

*where:*

$$M(\mathcal{K}) = \sum_{i=1}^{q}||\boldsymbol{K}_i||_p \ .$$

*Proof of Lemma S9.* Let $L = m + n + 2q - 2$. Starting from the definition of the ERP distance in Lemma S4 and the definition of the convolutional layer in Definition 4.2:

$$d_{\mathrm{ERP}}^p(C(F(P)), C(F(Q)))$$

$$= \min_{\substack{Z^a \in \mathcal{Z}_{m+q-1,L}, \\ Z^b \in \mathcal{Z}_{n+q-1,L}}} \sum_{k=1}^{L} \left|\left| z_k^a(C(F(P))) - z_k^b(C(F(Q))) \right|\right|_p$$

$$= \min_{\substack{Z^a \in \mathcal{Z}_{m+q-1,L}, \\ Z^b \in \mathcal{Z}_{n+q-1,L}}} \sum_{k=1}^{L} \left|\left| z_k^a \left( \left[ \sum_{j=1}^{m} \hat{K}_{i,j} \hat{f}_j(P) \right]_{i=1}^{m+q-1} \right) - z_k^b \left( \left[ \sum_{l=1}^{n} \hat{K}_{i,l} \hat{f}_l(Q) \right]_{i=1}^{n+q-1} \right) \right|\right|_p$$

$$= \min_{\substack{Z^a \in \mathcal{Z}_{m+q-1,L}, \\ Z^b \in \mathcal{Z}_{n+q-1,L}}} \sum_{k=1}^{L} \left|\left| z_k^a \left( \left[ \sum_{j=1}^{q} K_j f_{i+j-1}(P) \right]_{i=1}^{m+q-1} \right) - z_k^b \left( \left[ \sum_{j=1}^{q} K_j f_{i+j-1}(Q) \right]_{i=1}^{n+q-1} \right) \right|\right|_p$$

$$= \min_{\substack{Z^a \in \mathcal{Z}_{m+q-1,L}, \\ Z^b \in \mathcal{Z}_{n+q-1,L}}} \sum_{k=1}^{L} \left|\left| \sum_{j=1}^{q} K_j \left( z_k^a \left( [f_{i+j-1}(P)]_{i=1}^{m+q-1} \right) - z_k^b \left( [f_{i+j-1}(Q)]_{i=1}^{n+q-1} \right) \right) \right|\right|_p$$

$$\leq \min_{\substack{Z^a \in \mathcal{Z}_{m+q-1,L}, \\ Z^b \in \mathcal{Z}_{n+q-1,L}}} \sum_{k=1}^{L} \sum_{j=1}^{q} ||K_j||_p \cdot \left|\left| z_k^a \left( [f_{i+j-1}(P)]_{i=1}^{m+q-1} \right) - z_k^b \left( [f_{i+j-1}(Q)]_{i=1}^{n+q-1} \right) \right|\right|_p$$

$$= \min_{\substack{Z^a \in \mathcal{Z}_{m+q-1,L}, \\ Z^b \in \mathcal{Z}_{n+q-1,L}}} \sum_{j=1}^{q} ||K_j||_p \cdot \sum_{k=1}^{L} \left|\left| z_k^a \left( [f_{i+j-1}(P)]_{i=1}^{m+q-1} \right) - z_k^b \left( [f_{i+j-1}(Q)]_{i=1}^{n+q-1} \right) \right|\right|_p$$

$$= \sum_{j=1}^{q} ||K_j||_p \cdot d_{\mathrm{ERP}}^p (F(P), F(Q)) ,$$

where the last equality follows from the fact that $[f_{i+j-1}(P)]_{i=1}^{m+q-1}$ and $[f_{i+j-1}(Q)]_{i=1}^{n+q-1}$ are just windows of $F(P)$ and $F(Q)$ respectively including the complete sequences $F(P)$ and $F(Q)$, resulting in $d_{\mathrm{ERP}}^p \left( [f_{i+j-1}(P)]_{i=1}^{m+q-1}, [f_{i+j-1}(Q)]_{i=1}^{n+q-1} \right) = d_{\mathrm{ERP}}^p (F(P), F(Q)) \quad \forall j = 1, \cdots, q$. $\quad\square$

*Proof of Theorem 4.3.* We will bound the absolute value of the difference of outputs for two sentences $P, Q \in \mathcal{S}(\Gamma)$ of lengths $m$ and $n$ respectively. For any $y$ and $\hat{y}$:

$$|g_{y,\hat{y}}(P) - g_{y,\hat{y}}(Q)| := \left| \left( \sum_{i=1}^{m+l\cdot(q-1)} \sigma \left( c_i^{(l)}(PE) \right) - \sum_{j=1}^{n+l\cdot(q-1)} \sigma \left( c_j^{(l)}(QE) \right) \right) (w_{\hat{y}} - w_y) \right|$$

$$\text{[Hölder's inequality]} \leq ||w_{\hat{y}} - w_y||_r \cdot \left|\left| \sum_{i=1}^{m+l\cdot(q-1)} \sigma \left( c_i^{(l)}(PE) \right) - \sum_{j=1}^{n+l\cdot(q-1)} \sigma \left( c_j^{(l)}(QE) \right) \right|\right|_p$$

$$\text{[Lemma S5]} \leq ||w_{\hat{y}} - w_y||_r \cdot d_{\mathrm{ERP}}^p \left( \sigma \left( C^{(l)}(PE) \right), \sigma \left( C^{(l)}(QE) \right) \right)$$

$$\text{[Lemma S8 and Lemma S9 recursively]} \leq ||w_{\hat{y}} - w_y||_r \cdot \left( \prod_{k=1}^{l} M(\mathcal{K}^{(k)}) \right) \cdot d_{\mathrm{ERP}}^p (PE, QE)$$

$$\text{[Lemma S7]} \leq ||w_{\hat{y}} - w_y||_r \cdot \left( \prod_{k=1}^{l} M(\mathcal{K}^{(k)}) \right) \cdot M(E) \cdot d_{\mathrm{Lev}}(P, Q) .$$

$\quad\square$

## B.1 REMARKS REGARDING GLOBAL AND LOCAL LIPSCHITZ CONSTANTS

In this section we introduce some interesting remarks regarding how global and local Lipschitz constants are employed in this work. We define global and local Lipschitz constants as:

**Definition S10** (Global Lipschitz constant). Let $g : \mathcal{S}(\Gamma) \to \mathbb{R}$ and $d_{\text{Lev}}$ be defined as in Section 3. We say $G$ is a global Lipschitz constant if:

$$|g(\boldsymbol{P}) - g(\boldsymbol{Q})| \leq G \cdot d_{\text{Lev}}(\boldsymbol{P}, \boldsymbol{Q}) \ \ \forall \boldsymbol{P}, \boldsymbol{Q} \in \mathcal{S}(\Gamma) \,.$$

The Lipschitz constant $G$ in Definition S10 is valid for any two sentences $\boldsymbol{P}$ and $\boldsymbol{Q}$, one example is our Lipschitz constant in Theorem 4.3. When additional conditions are posed on the set of sentences where $G$ is valid, we say a Lipschitz constant is *local*. A specific case case of locality is when the Lipschitz constant depends on one of the arguments of the distance as $G(\boldsymbol{P})$, this is the kind of local Lipschitz constants we observe in this work:

**Definition S11** (Local Lipschitz constant). Let $g : \mathcal{S}(\Gamma) \to \mathbb{R}$ and $d_{\text{Lev}}$ be defined as in Section 3. We say $G : \mathcal{S}(\Gamma) \to \mathbb{R}^+$ is a global Lipschitz constant if:

$$|g(\boldsymbol{P}) - g(\boldsymbol{Q})| \leq G(\boldsymbol{P}) \cdot d_{\text{Lev}}(\boldsymbol{P}, \boldsymbol{Q}) \ \ \forall \boldsymbol{P}, \boldsymbol{Q} \in \mathcal{S}(\Gamma) \,.$$

Some examples of such local Lipschitz constants are Remark 4.5 and Corollary 4.6. Some properties to consider regarding global and local Lipschitz constants are:

*Remark* S12 (Global Lipschitz constants upper bound local Lipschitz constants). Let $G$ be a global Lipschitz constant as in Definition S10 and $G : \mathcal{S}(\Gamma) \to \mathbb{R}^+$ a local Lipschitz constant as in Definition S11, we have that:

$$G(\boldsymbol{P}) \leq G \quad \forall \boldsymbol{P} \in \mathcal{S}(\Gamma) \,.$$

*Remark* S13 (Local Lipschitz constants might not hold everywhere). Let $G : \mathcal{S}(\Gamma) \to \mathbb{R}^+$ be a local Lipschitz constant as in Definition S11, there might exist some $\boldsymbol{P}, \boldsymbol{Q}, \boldsymbol{R} \in \mathcal{S}(\Gamma)$ such that:

$$|g(\boldsymbol{Q}) - g(\boldsymbol{R})| > G(\boldsymbol{P}) \cdot d_{\text{Lev}}(\boldsymbol{Q}, \boldsymbol{R}) \,.$$

Remarks S12 and S13 highlight the two key aspects of local Lipschitz constants. While the bound is tighter than for global Lipschitz constants (Remark S12), these bounds can only be employed to give the certified radius around the sentence $\boldsymbol{P}$ where we compute the local Lipschitz constant $G(\boldsymbol{P})$, otherwise, the Lipschitzness property is lost (Remark S13).

