# OpenReview forum: "Certified Robustness Under Bounded Levenshtein Distance"
_ICLR.cc/2025/Conference — ICLR 2025 Poster_

### Official Review · Reviewer_PA7T · 2024-10-29

**Soundness:** 3
**Presentation:** 2
**Contribution:** 2
**Rating:** 6
**Confidence:** 4

**Summary:**

This paper derives the Lipschitz constant of convolutional networks with respect to the Levenschtein distance on its textual inputs. As a result, certified radii (in the Levenschtein distance) for robustness are provided. It is shown how to use the derived Lipschitz bound in order to train a 1-Lipschitz model (with respect to the Levenschtein distance). Furthermore, experiments are conducted to validate the effectiveness of the proposed robustness certificates.

**Strengths:**

The topic is certainly of interest to the ICLR community. The mathematical derivations appear correct. The proposed ideas are novel and well-motivated.

**Weaknesses:**

I have concerns regarding the presentation and performance limitations of the proposed method. See my Questions below for more details.

**Questions:**

1. Line 161: How is $d_\text{edit}$ defined? It would be good to explicitly define this for readers that are from outside the NLP community. It would also be good to clarify exactly what the embedding matrix $\mathbf{E}$ is.
2. Line 163: In the introduction, you said that you would denote the elements of a matrix by $x_{ij}$ (no comma in the subscript). However, you're now using commas in the subscripts (e.g., $z_{i,j}$ and $e_{k,i}$). It would be good to fix things to use consistent notation throughout the paper.
3. Line 173: "tenths". It looks like you probably meant "tens of thousands."
4. In Section 4.1, you are concerned with local Lipschitz constants, but then in Section 4.2 you begin to discuss global Lipschitz continuity. Why the discrepancy?
5. Line 253: The footnote 1 should be put after the punctuation. Otherwise, as it is now, it looks like it is part of the mathematical expression (i.e., it looks like "one to the power of one").
6. Remark 4.5: Could you explain how this is a local Lipschitz constant? This seems like a global bound if it indeed holds for all $\mathbf{P}$ and $\mathbf{Q}$.
7. Table 2 makes your method look quite loose, with IBP and BruteF attaining higher verified accuracies than yours in most of the $k=1$ settings. I understand that the other methods are more computationally expensive/are not applicable to $k=2$, but the results being displayed here are not very convincing.
8. "Our method, LipsLev, is able to provide the best performance in AG-News" Isn't BruteF providing the best performance for AG-News?

---

> ### Author Response · Authors · 2024-11-18
>
> Dear Reviewer PA7T,
>
> Thank you for your thorough review. Your comments have been incorporated to the text and helped us improve the quality of our work. Please find our answer to your comments bellow:
>
> - **P1: How is $d_{\text{edit}}$ defined?**
>
> The edit distance without insertions and deletions for two sentences of one-hot vectors of same length $\mathbf{P} \in \{0,1\}^{m\times d}$ and $\mathbf{Q} \in \{0,1\}^{m\times d}$ is defined as:
>
> $$
> d_{\text{edit}}(\mathbf{P}, \mathbf{Q}) = \sum_{i=1}^{m} \|\|(\mathbf{p}_i - \mathbf{q}_i)\|\| _{\infty}
> $$
>
> i.e., the number of positions in the sentence where the characters are different. We have added this definition in lines 160-161.
>
> - **P2: What is exactly the embedding matrix $\mathbf{E}$?**
>
> As mentioned in line 210 of the original submission, $\mathbf{E} \in \mathbb{R}^{|\Gamma|\times d}$ is simply a real matrix that is learnt during training. Each row of $\mathbf{E}$ contains a real vector representing each character. This represents what the torch layer [torch.nn.Embedding](https://pytorch.org/docs/stable/generated/torch.nn.Embedding.html) does.
>
> - **P3: Typos:**
>
>     - Mismatching notation in line 163
>
>     We thank the reviewer for pointing out this inconsistency, we have removed the commas.
>
>     - Typo in line 173: "tenths" -> "tens"
>
>     Thanks for pointing this out, it should be "tens" and has been corrected.
>
>     - Footnote 1 should be put after the punctuation
>
>     Thanks for the suggestion. We have corrected it.
>
> - **P4: In section 4.1 you are interested in local Lipschitz constants, but then in 4.2 you discuss global ones.**
>
> That is true, in the end, in Corollary 4.6 we present a local Lipschitz constant, but the introductory analysis in section 4.1 is with global Lipschitz constants. We have corrected the typo and put "global" instead of "local" in section 4.1.
>
> - **P5: Why is Remark 4.5 a local Lipschitz constant for the embedding layer?**
>
> We have added a discussion on this topic in Appendix B.1 and the [General response](https://openreview.net/forum?id=cd79pbXi4N&noteId=hLsRhMYK2m).
>
> - **P6: Your method is much more efficient, but the verified accuracy is worse than IBP and BruteForce in some cases for $k=1$.**
>
> That is true. However, we believe that efficiency is a more important factor to consider. The brute force approach should only be consider as a baseline to see how far is a method from perfect verified accuracy and how much faster it is. In this sense, the IBP approach of [1] with Levenshtein distance constraints serves little purpose. IBP attains a smaller verified accuracy than IPB with a larger runtime. The brute force runtime should be an upper bound for any verification method. The benefit of LipsLev comes from its efficiency. Moreover, the fact that it can match or surpass IBP in some cases is an surplus.
>
> - **P7: Regarding lines 402-403. Isn't BruteF providing the best performance for AG-News?**
>
> That is true. In lines 402-403 of the original manuscript we wanted to highlight that LipsLev is better than IBP. As covered in the previous point and in lines 365-368 of the original manuscript, no method can be better than the brute force approach. We have updated the writing to make clear than LipsLev is better than IBP, not Brute Force.
>
> We will be happy to answer to any additional concerns you have. If your concerns are addressed, please, consider increasing the score.
>
> **References**
>
> [1] Huang et al., Achieving verified robustness to symbol substitutions via interval bound propagation, EMNLP 2019

---

> ### Comment · Reviewer_PA7T · 2024-11-19
>
> I thank the authors for their thoughtful responses and edits to the manuscript. I have increased my overall Rating by 1 point.

---

> > ### Author Response · Authors · 2024-11-19
> > **Thanks for your response**
> >
> > Dear Reviewer PA7T,
> >
> > We appreciate the acknowledgement of our rebuttal and the increase in the score. Please do not hesitate to reach us with any additional questions. We remain available until the discussion window closes.
> >
> > Regards,
> >
> > Authors

---

### Official Review · Reviewer_U4sY · 2024-10-31

**Soundness:** 4
**Presentation:** 3
**Contribution:** 2
**Rating:** 8
**Confidence:** 3

**Summary:**

This work presents a convolutional text classifier which is certifiably robust to Levenshtein-distance perturbations.

**Strengths:**

Overall, this work is a solid academic contribution, albeit with limited prospects for scalability.

1. I like that this approach provides closed-form, deterministic certificates. It is the first paper that I have seen tackling the challenging Levenstein-distance setting without resorting to randomized approaches.
2. The paper writing is clean and structurally sensible. The problem setting is well-motivated and the necessary prior work is precisely and clearly introduced.
3. Figure and table presentation is thoughtful -- I appreciate the coordinated highlighting between Table 2 and Table 3.
4. Method runtime is orders of magnitude faster than baselines; this is expected as the Lipschitzness constant is enforced by construction.
5. The theory is carefully presented and a quick scan of the proof strategy in appendix B makes sense, although I haven't checked the proofs in detail.

**Weaknesses:**

1. My main concern with this approach is that a) the certified edit distances are very small (one or two characters), b) the models are tiny (only one convolutional layer), and c) I think possibility for future improvement is minimal. Layer-level Lipschitz constants have been more or less discarded by the image classification robustness community for several years now, as exponential signal attenuation makes it difficult to construct verifiable deep networks. This problem setting is arguably much more difficult, and since layer-wise Lipschitz approaches have proven intractable for images, I don't expect that this approach will be scalable to any reasonable network size.
2. Section 6 discusses extensions to transformers, which is too far removed from what is presented in the paper. I understand the author's desire to connect their approach with SOTA language models, but a one-layer convolutional classifier is nowhere close to the complexity of a transformer.
3. The literature review should be a little more comprehensive, and do a better job of grouping related works. For example, a quick google search for "certifiable robustness text classifier" yields [1], which is certainly related work.
4. The authors don't evaluate against randomized methods. While I understand that deterministic methods are a more apples-to-apples comparison, some readers might appreciate fully comprehensive baselines (perhaps in the appendix).

[1] Siqi Sun and Wenjie Ruan. 2023. TextVerifier: Robustness Verification for Textual Classifiers with Certifiable Guarantees. In Findings of the Association for Computational Linguistics: ACL 2023, pages 4362–4380, Toronto, Canada. Association for Computational Linguistics

**Questions:**

1. To confirm: Theorem 4.3 holds for any choice of $p$ and $r$ which satisfy $1/p + 1/r = 1$?
2. A suggestion for Figure 1: I think this would be better presented a line graph with the y axis as "percent of sentences verified." That's what you're actually trying to communicate: that as the sentence length goes up, a high fraction are verified. Right now, readers have to mentally "divide the columns" to  back out what you're trying to say. Another alternative is a stacked bar chart based on proportions, e.g. something like [this](https://community.tableau.com/sfc/servlet.shepherd/version/renditionDownload?rendition=THUMB720BY480&versionId=0684T000002JELx&operationContext=CHATTER&contentId=05T4T000008xZ3u&page=0).
3. Have the authors empirically verified their certificates by running inference on random string modifications within the Levenstein distance of the certificate? Just to check that there's no mistake in the proofs that is slipping through.
4. Related to the previous point, I'd be curious to see the gap between the empirical and theoretical certificates, perhaps adapting a greedy coordinate-wise attacking strategy as in [1]. But I understand that this is orthogonal to the certified robustness focus of the paper, so this is just a suggestion to the authors if they are independently compelled to investigate.

Minor notes:
1. "English" on line 92 should be capitalized.
2. In Definition 4.1, it seems that the notation of $d_{ERP}$ should also be dependent on $p$? I.e. $d_{ERP}^p(A,B)$.

[1] Zou, Andy, et al. "Universal and transferable adversarial attacks on aligned language models." arXiv preprint arXiv:2307.15043 (2023).

---

> ### Author Response · Authors · 2024-11-18
>
> Dear Reviewer U4sY,
>
> We are thankful for the nice comments and the thorough review. Please find the responses to your comments bellow:
>
> - **P1: "I think possibility for future improvement is minimal ... Layer-level Lipschitz constants have been more or less discarded by the image classification robustness community for several years now, as exponential signal attenuation makes it difficult to construct verifiable deep networks".**
>
> We kindly disagree with the reviewer. To the best of our knowledge, Lipschitz networks in Computer Vision are actively being developed [2,3,4], with 1 submission to this ICLR 2025 [2]. The "signal attenuation" problem has been addressed in CV by proposing norm preserving (orthogonal) layers [5], with existing works training up to 12-layer networks [3,4]. Such results do not extend naturally to the NLP setup as existing methods treat $\ell_p$ norms. We conjecture that in our case, the sum of the $\ell_p$ norm of each vector in the sequence should be preserved.
>
> It is true that the state-of-the art in verification in NLP is far behind CV and there is a lot to do. Nevertheless, we believe that our work opens up many interesting challenges as developing norm-preserving layers or the challenges discussed in lines 464-483.
>
> - **P2: Section 6 discusses challenges to verify transformers, but the models studied in the paper are far behind transformers.**
>
> Indeed, the architectures studied in our paper are nowhere near a production LLM. However, we believe our results are promising and the extension to encoder-only transformers for text classification is not far-fetched. We think that highlighting the challenges in Section 6 can be interesting for the community and useful for achieving this goal.
>
> - **P3: Missing related work.**
>
> Thanks for providing this reference, it is indeed relevant and it has been added in Lines 99-100.
>
> - **P4: Empirical comparison against randomized methods.**
>
> Thanks for the suggestion. As you mentioned, comparing sound verification methods with probabilistic ones is not completely fair. Nevertheless we think it is a good idea and we are working on testing RSDel in our setup. Unfortunately, we will not be able to provide the results before the discussion window closes. The results will be added in the appendix of the camera ready version.
>
> - **P5: Theorem 4.3 holds for any choice of $p$ and $r$ that satisfy $1/p + 1/r = 1$?**
>
> Thanks for highlighting this detail. Yes, but also we need $p\geq 1$. This is obvious as $p$ and $r$ should induce a $p$-norm. We have clarified this detail in the definition of the ERP distance and in Theorem 4.3.
>
> - **P6: The message of Figure 1 could be expressed more effectively.**
>
> Thanks for this suggestion, we have included the verified accuracy v.s. sentence length in Figure 1. For completeness, we have moved our original plots with the full distribution to Appendix A.2. With the new version, it is more clear that longer sentences are easier to verify.
>
> - **P7: Have authors tried computing the empirical robustness via adversarial attacks?**
>
> We are thankful to the reviewer for this suggestion. We think this experiment adds value to the paper. Please check our [General response](https://openreview.net/forum?id=cd79pbXi4N&noteId=hLsRhMYK2m) for a discussion.
>
> - **P8: English should be capitalized on line 92.**
>
> Sorry for the mistake and thanks for pointing it out. It has been corrected.
>
> - **P9: $d_{\text{ERP}}$ depends on $p$, it could be incorporated in the notation as $d^{p}_{\text{ERP}}$**
>
> Indeed, we think it is a good idea. We thank the reviewer for the suggestion. We have updated the notation in the text.
>
> We remain available for further discussion. Please let us know if you have additional concerns. If you are satisfied, we would appreciate an increase in the rating.
>
> **References**
>
> [1] Abad Rocamora et al., Revisiting Character-level adversarial attacks for Language Models, ICML 2024
>
> [2] Anonymous, Enhancing Certified Robustness via Block Reflector Orthogonal Layers, ICLR 2025 submission
>
> [3] Hu et al., Unlocking Deterministic Robustness Certification on ImageNet, NeurIPS 2023
>
> [4] Hu et al., A recipe for improved certifiable robustness, ICLR 2024
>
> [5] Li et al., Preventing Gradient Attenuation in Lipschitz Constrained Convolutional Networks, NeurIPS 2019

---

> > ### Comment · Reviewer_U4sY · 2024-11-19
> > **Thank you for your response**
> >
> > I do not fully agree with the authors on the first point; if [5] had fully addressed the attenuation problem in 2019, we would have seen Lipschitz classifiers with strong clean and robust performance ([3] only achieves ~81% on CIFAR-10). Nevertheless, I think this is sitll a solid contribution, and I am willing to raise my score.

---

> > > ### Author Response · Authors · 2024-11-20
> > > **Thanks for your response**
> > >
> > > Dear reviewer U4sY,
> > >
> > > Thanks for acknowledging our rebuttal and increasing the score. Your comments helped us improve the quality of our work.
> > >
> > > Regarding the attenuation problem, we agree that the attenuation problem is not fully solved, with many works on the topic after 2019, even submitted to this conference [2]. The message we wanted to convey is that orthogonal layers could be devised as well in our setup in order to scale up to deeper models.
> > >
> > > As per the performance, we do not think that fully solving attenuation can match the clean accuracy of standard models. Enforcing 1-Lipschitzness significantly constrains the model. Solving attenuation will allow optimizing better these models, but the maximum performance they can get is still going to be constrained.
> > >
> > > Thanks for the interesting discussion, regards,
> > >
> > > Authors

---

### Official Review · Reviewer_Mjg7 · 2024-11-03

**Soundness:** 3
**Presentation:** 3
**Contribution:** 3
**Rating:** 6
**Confidence:** 3

**Summary:**

This paper studies the certified robustness of convolutional neural networks for text classification under Levenshtein distance. The major challenge of this setting compared with standard certified robustness is twofold: one is the non-fixed length of text under perturbation, and the other is the discrete nature of tokens. The authors define the concept the Lipschitzness in this setting in terms of Levenshtein distance, and derived a relxation method for compute the Lipschitzness. The approach can be readily used to train certifiably robust models. Results show that the method is efficient and can achieve a performance on par with previous IBP-based certification and extend to larger perturbation scenarios.

**Strengths:**

1. The problem is significant and challenging. The original problem of certified robustness typically focuses on continuous, fixed-demension inputs, which is different from real-world NLP setting. The authors instead considered Levenshtein distance, which is a reasonable metric.
2. The proposed solution is interesting, efficient, and simple. This is a major advantage of this paper compared with prior approaches.
3. The writing is quite clear and readable. I enjoy reading this paper a lot.

**Weaknesses:**

1. One major problem is the Lipschitzness constraint is so strong and perhaps harms the expressive power of resulting model, thus hampering the clean accuracy. In the standard certified robustness community in CV, constraining the Lipschitz constant does not show performance comparable to IBP. This problem is also theoretically studied in [Zhang et al. 2022]. Do you think your models in experiment suffers from this problem in clean accuracy compared with other approaches in small perturbation scenario?

Reference: [Zhang et al. 2022] Rethinking lipschitz neural networks and certified robustness: A boolean function perspective. NeurIPS 2022.

2. Regarding the experimental results, the performance where Levenshtein distance is 1 does not show the advantage of the proposed method. Although the training and verification is quite efficient, the performance on SST-2 and IMDB is not comparable with the simplest IBP.

**Questions:**

1. Why on FakeNews dataset LipsLev has exactly the same performance as IBP when the Levenshtein distance is 1?

Miscellaneous:
1. Line 108: $s_i$ should be $S_i$?
2. Line 240: $\mathbb R^{m+(q-1)\times d}$ should be $\mathbb R^{m+2(q-1)\times d}$?

---

> ### Author Response · Authors · 2024-11-18
>
> Dear Reviewer Mjg7,
>
> Thank you for your review. We are happy to hear that you enjoyed reading our paper. We comment on your points bellow:
>
> - **P1: Constraining the Lipschitz constant in Computer Vision models can harm the expressiveness of the model. Is it the case in your setup?**
>
> That is also the case in our setup. However, having a small Lipschitz constant is necessary both for robustness and verifiability. Please check the results in Table 3 regarding Lipschitz regularization. Models with a lower regularization weight for the Lipschitz constant achieve a higher clean accuracy, but near-zero verified accuracy. When a higher regularization weight is employed, we obtain larger verified accuracies but clean accuracy is hindered. We have linked our results with the insights from [2] in $\ell_p$ spaces in Section 5.3.
>
> - **P2: The performance at distance 1 is not better than IBP in some cases.**
>
> That is true, IBP [1] is a very expensive approach that can obtain high verified accuracies. However, the runtime is larger and performance is lower than the brute force approach, which provides the exact robustness of the model. We believe that it is impractical to employ either brute force or the IBP approach of [1] in our setup. Nevertheless, the fact that LipsLev can sometimes beat IBP is a good indicator of the power of our approach.
>
> - **P3: Why does LipsLev have the same performance as IBP on FakeNews?**
>
> This is due to pure chance. As mentioned in lines 324-326, due to the extreme verification costs of the brute force and IBP approaches, we limit the samples used for FakeNews to $50$. It is likely that the performance would change with more samples. In Table S5, we extended the FakeNews evaluation with LipsLev to $1000$ samples and $k=10$ for completeness.
>
> - **P4: Line 108: should $\mathbf{s}_i$ be $\mathbf{S}_i$?**
>
> No, according to our notation convention at the end of the Introduction, we use lowercase bold letters for vectors, since $\mathbf{s}_i$ is the i$^{th}$ row of $\mathbf{S}$, it should be lowercase.
>
> - **P5: Line 240: should $\mathbf{R}^{m + (q-1)\times d}$ be $\mathbf{R}^{m + 2\cdot(q-1)\times d}$?**
>
> Yes, thanks for pointing out this typo, it has been corrected.
>
> We will be happy to discuss any other aspects you consider relevant and remain available until the discussion window closes. If you are happy with our changes and clarifications, we would appreciate a raise in the score.
>
> **References**
>
> [1] Huang et al., Achieving verified robustness to symbol substitutions via interval bound propagation, EMNLP 2019
>
> [2] Zhang et al., Rethinking Lipschitz Neural Networks and Certified Robustness: A Boolean Function Perspective, NeurIPS 2022

---

> > ### Author Response · Authors · 2024-11-25
> > **Have we addressed your concerns?**
> >
> > Dear reviewer Mjg7,
> >
> > Thanks again for your review. Since the discussion window is coming to an end, we wanted to check whether our rebuttal addressed your concerns and if you have any additional questions.
> >
> > Regarding your third point in our rebuttal, we are running 50 extra samples in the Fake-News dataset to have a more accurate estimate of the performance of each method. Unfortunately, due to the extreme time consumption of BruteF and IBP, we do not have the numbers yet, but they will be included in the camera ready version.
> >
> > Best regards,
> >
> > Authors

---

> > > ### Comment · Reviewer_Mjg7 · 2024-11-25
> > > **Thank you**
> > >
> > > Thank you for your response. My question 1 has been addressed. I generally feel that this paper points out a promising direction, although the performance is still not so exciting. Regarding my concern of global Lipschitzness, I still think the condition may be too strong when the perturbation is not that large (i.e., distance <= 1). For CV robustness, it is already a commonsense that Lipschitz models cannot outperform IBP or other local certification methods when eps is small (e.g., 2/255 on CIFAR-10), but it outperforms IBP when eps is large (e.g., 8/255). Overall, I think designing global Lipschitz models may not be the optimal method for dealing with your setting (distance=1), but the proposed direction may be promising for larger perturbations. I thus tend towards maintaining my score of acceptance.

---

> > > > ### Author Response · Authors · 2024-11-25
> > > > **Thanks for your response**
> > > >
> > > > Thanks for replying to our rebuttal.
> > > >
> > > > We agree with the reviewer in that for smaller distances in CV, verification methods like IBP or MILP solvers are usually better than Lipschitz-based verifiers. We would like to clarify that one key difference is that the complexity of IBP in CV is similar to a single forward pass. In our setup, IBP requires many forward passes and can only verify at distance 1.
> > > >
> > > > Additionally, please note that **our setting is not distance 1**, with our method, we verify at distance 2 in Table 2 for all models and datasets. Moreover, we can obtain non-zero verified accuracy at distance up to $10$ in the Fake-News dataset, see Table S5.
> > > >
> > > > Regards,
> > > >
> > > > Authors

---

> > > > > ### Author Response · Authors · 2024-12-02
> > > > > **Have we addressed your concerns?**
> > > > >
> > > > > Dear reviewer Mjg7,
> > > > >
> > > > > We have completed the additional $50$ points for validating the verified accuracies in Fake-News regarding your point **P3** in our first response. We could only run $50$ extra points as the brute force and IBP approaches were taking around 17 min per sentence.
> > > > >
> > > > > **Fake-News verified accuracy**
> > > > > |$p$ | Acc. | $k$ | Charmer | BruteF | IBP | LipsLev (Ours) |
> > > > > | --- | --- | --- | --- | --- | --- | --- |
> > > > > |$\infty$| $94.33_{\pm (0.94)}$ | $1$ | $86.67_{\pm (0.94)}$ | $84.67_{\pm (0.94)}$ | $83.00_{\pm (0.82)}$ | $83.00_{\pm (0.82)}$ |
> > > > > |$\infty$| $94.33_{\pm (0.94)}$ | $2$ | $76.00_{\pm (1.63)}$ | OOT | X | $71.00_{\pm (0.82)}$ |
> > > > > |$1$| $94.33_{\pm (0.47)}$ | $1$ |$92.00_{\pm (0.00)}$ | $88.67_{\pm (0.47)}$ | $87.00_{\pm (0.82)}$ | $87.00_{\pm (1.41)}$ |
> > > > > |$1$| $94.33_{\pm (0.47)}$ | $2$ | $79.33_{\pm (2.49)}$ | OOT | X | $74.67_{\pm (2.87)}$ |
> > > > > |$2$| $94.00_{\pm (0.82)}$ | $1$ | $89.00_{\pm (4.55)}$ | $86.67_{\pm (2.87)}$ | $85.00_{\pm (1.63)}$ | $85.33_{\pm (3.86)}$ |
> > > > > |$2$| $94.00_{\pm (0.82)}$ | $2$ | $78.00_{\pm (4.32)}$ | OOT | X | $70.67_{\pm (5.44)}$ |
> > > > >
> > > > > In the table, we can observe that the performance of IBP and LipsLev is not exactly the same, with the standard deviation being different for $p=1$ and LipsLev having $0.33$\% points more than IBP for $p=2$. This is consistent with our prior intuition that the two methods having the same performance was a coincidence. If your concerns are satisfied, we would apreciate an increase in the score. Please let us know if anything remains unclear.
> > > > >
> > > > > Regards,
> > > > >
> > > > > Authors

---

### Official Review · Reviewer_39zs · 2024-11-04

**Soundness:** 3
**Presentation:** 3
**Contribution:** 3
**Rating:** 8
**Confidence:** 3

**Summary:**

The paper introduces a technique to verify the robustness of a neural network for text classification tasks in a natural language processing setting (NLP).
In NLP, the Levenshtein distance is a natural way to measure the distance between two strings.
Existing verification approaches, that often rely on interval bound propagation (IBP), are however prohibitively expensive when using the Levenshtein distance to measure robustness radia.
To address this, the authors present a Lipschitz constant-based verification technique that allows to compute a certified robustness radius for convolutional models, once the Lipschitz constant of the classifier is known.
For their verification procedure, the authors consider a real-valued extension of the Levenshtein distance and a specific convolutional architecture, for which they derive robustness radius bounds that can be computed with a single forward pass.
Compared to existing IBP and brute force approaches, the proposed method is significantly faster and provides comparable clean/verified accuracy ratio for a Levenshtein distance of 1.
Remarkably, the proposed method is the only one which can provide robustness guarantees for a Levenshtein distance of 2.

**Strengths:**

The authors address an interesting problem in NLP, which is of great relevance given the prevalence, and related risks, of large language models.
From a wider perspective, it is important to devise techniques to provide robustness certifications which are practically computable.
The paper has thus a strong motivation.
With respect to the approach, the paper clearly presents the limitations of existing techniques and orderly discusses the proposed solution.
The paper is well structured, and it seems sound and of potential practical relevance to me.

**Weaknesses:**

**Relevance of ReLU-activated convolutional networks**

While the paper is generally well motivated and an architecture similar to the one discussed has been previously investigated, I think that further details on the relevance of the method presented would be helpful.
Specifically, it is not clear how practically useful ReLU-activated, fully-connected convolutional networks are in text classification.
As part of the motivation is to make NLP verification practically achievable, I think that a more detailed discussion of the limitations of the proposed method should be part of the main body of the paper.


**Definition of terms**

Some notions and terminology used require more explicit definition.
Firstly, Lipschitz continuity, as well as related concepts such as the "local Lipschitz constant", is never explicitly introduced.
As the paper heavily relies on these concepts, they should be introduced.
Similarly, the notions of "clean accuracy" and "verified accuracy" are not clearly defined.
As these are used as the main performance metrics, they should be properly described as well.



**Empirical evaluation**

Related to my previous comment, further explanations on the behaviour of the clean accuracy for networks with more than one layer is necessary.
In figure S4, the clean accuracy drops significantly as more than 2 layers are used: why is this behaviour observed?


**Minor comments**

* line 52, of with -> with
* (several places) Related works -> related work

**Questions:**

* What is the relevance of methods that combine several algorithms such as alpha-beta-CROWN in your investigation? Can such approaches be adapted to the NLP setting?

* The approach you propose is sound, can you discuss completeness too?

---

> ### Author Response · Authors · 2024-11-18
>
> Dear Reviewer 39zs,
>
> We are thankful for your feedback and appreciate the positive comments about the potential of our approach. We address your points as follows:
>
> - **P1: Can you discuss more on the limitations of your approach?**
>
> We refer the reviewer to lines 463-483 of the main text, where we discuss the challenges and limitations of our approach in the context of modern NLP. Concretely, we believe the main limitations are (i) The difficulty to train and verify deep models as shown in Appendix A.6. (ii) The current Lipschitz constant bounds do not apply to Transformer models.
>
>
> We would be happy to discuss on other limitations the reviewer might find relevant.
>
> - **P2: What is the difference between Local and Global Lipschitz constants?**
>
> We have clarified this aspect in Appendix B.1, please check the [General response](https://openreview.net/forum?id=cd79pbXi4N&noteId=hLsRhMYK2m) for a discussion.
>
>
> - **P3: Can you define Clean, Adversarial and Verified Accuracy?**
>
> Sure, the clean accuracy is simply the percentage of correctly classified points in a given dataset, no adversarial attacks involved. The adversarial accuracy is the percentage of points that are correctly classified after an attack has been performed. Finally, the verified accuracy is the percentage of points that we can verify the prediction is correct in a given radius. This has been defined more formally in Appendix A1.
>
> - **P4: Performance degrades with more than two layers. Can you comment on that?**
>
> We believe that when imposing 1-Lipschitzness with multiple layers, we incur in "Gradient attenuation", which has been observed in CV for $\ell_p$ spaces [1]. It remains an open problem to address this issue in NLP.
>
> - **P5: Typos: "of with" -> "with", "related works" -> "related work".**
>
> Thanks for pointing out these typos, they have been corrected.
>
> - **P6: Can you adapt $\alpha$-$\beta$-CROWN or other verification methods in computer vision to your setup?**
>
> Overall, Bound propagation methods like $\alpha$-$\beta$-CROWN rely on linear upper and lower bounds of each layer. These bounds are propagated through the model until the last layer in order to obtain linear upper and lower bounds of the output with respect to the input.
>
> One key problem that prevents from using these approaches is the changing lengths in NLP. If the number of neurons per layer changes with the input length, the linear upper and lower bounds are not well defined. For this same reason, existing IBP methods in NLP can only work with Levenshtein distance constraints if the bounds are computed after the pooling layer, where the number of neurons is constant.
>
> One key component in $\alpha$-$\beta$-CROWN that could be incorporated in NLP is branch and bound strategies, we discuss about this in the following point.
>
> - **P7: Your verification approach is sound. Is it complete?**
>
> In practice, for a verification method to be sound, it should not give false positives, where positive is verified and negative is falsified. For a verification method to be complete, it should not return false negatives. As displayed in our experiments, our method is not complete as it returns false negatives, i.e., we cannot verify some samples that the brute force method does.
>
> Completeness is often achieved by combining a sound method with a branch and bound strategy, which makes the bounds tighter by considering progressively smaller regions for verification [2]. We believe it is a very interesting avenue to explore such approaches in NLP, as, to the best of our knowledge, no method other than the BruteForce approach is sound and complete for Levenshtein distance constraints.
>
> We remain available for discussion and would be happy to answer to further questions.
>
> **References**
>
> [1] Li et al., Preventing Gradient Attenuation in Lipschitz Constrained Convolutional Networks, NeurIPS 2019
>
> [2] Bunel et al., Branch and Bound for Piecewise Linear Neural Network Verification, JMLR 2020

---

> > ### Comment · Reviewer_39zs · 2024-11-19
> >
> > Thank you very much for your answers and for updating the manuscript and improving its clarity. I will keep my score.

---

> ### Author Response · Authors · 2024-11-19
> **Thanks for your response**
>
> Dear Reviewer 39zs,
>
> Thank you for the acknowledgement of our rebuttal. Please let us know if additional questions come up.
>
> Regards,
>
> Authors

---

### Author Response · Authors · 2024-11-18
**General response**

Dear reviewers,

We are thankful for the thorough reviews that helped us improved the quality of our work and for the positive comments. We are happy you enjoyed reading our work, recognized the efficiency of our approach and its potential. Here, we summarize our changes in the manuscript:

- **Empirical robustness evaluation with character-level attacks**

Following the suggestion of Reviewer U4sY, we have ran Charmer [1] with its standard hyperparameters, k=1 and k=2 for all datasets in Table 2. The results have been included in Table 2 and with standard deviations in Table S4. As expected, the Charmer adversarial accuracy is always above every verification method. We find that the gap between the Charmer adversarial accuracy and the LipsLev verified accuracy at $k=2$ is large for the AG-News, SST-2 and IMDB datasets. It is likely that this large gap is a combination of Charmer being a looser upper bound and LipsLev being a looser lower bound of the true rubust accuracy for $k=2$. Nevertheless, for the Fake-News dataset, the gap between the LipsLev verified accuracy and the Charmer verified accuracy is as close as $75.33$% v.s. $79.33$% for $p=1$.

- **Remarks regarding global and local Lipschitz constants**

Let us clarify the difference between global and local Lipschitz constants. We have included a detailed discussion on this topic in Appendix B.1. Here we present a small summary taking Remark 4.5 as an example.


Local Lipschitz constants are important as they can give tighter bounds when additional conditions are met. Note that the Lipschitz constant in Remark 4.5 ($M(\mathbf{E}, \mathbf{P})$) depends on $\mathbf{P}$. Therefore, even though the bound holds for any $\mathbf{P}, \mathbf{Q}$, the specific constant $M(\mathbf{E}, \mathbf{P})$ might not be valid for another pair of sentences $\mathbf{P}', \mathbf{Q}'$, i.e., it could exist some $\mathbf{P}', \mathbf{Q}'$ so that
$$
    d_{\text{ERP}}(\mathbf{P}'\mathbf{E}, \mathbf{Q}'\mathbf{E}) > M(\mathbf{E}, \mathbf{P})\cdot d_{\text{Lev}}(\mathbf{P}', \mathbf{Q}')\,.
$$
That is why $M(\mathbf{E})$ and $M(\mathbf{E}, \mathbf{P})$ are the global and local Lipschitz constants respectively, with $M(\mathbf{E}) \geq M(\mathbf{E}, \mathbf{P})$. While $M(\mathbf{E})$ is looser, it is valid for any $\mathbf{P}$. In practice, $M(\mathbf{E}, \mathbf{P})$ can be computed with little overhead while doing the forward pass of $\mathbf{P}$ through the model. Therefore, local Lipschitz constants of the model can be computed for every sentence in the dataset.

- **Definitions of Clean, Adversarial and Verified accuracy**

Following the suggestion of Reviewer 39zs, we have defined these three terms in Appendix A.1.

- **Correction of typos**

Reviewers pointed out some typos and notation ambiguities that have been corrected in the text.

We have included all of the differences between the original submission and the rebuttal updates in blue color to ease up the reading. We remain available for discussion in case reviewers have remaining concerns.

---

### Meta-Review · Area_Chair_mA6j · 2024-12-21

**Metareview:**

This work proposes a convolutional text classifier which can provide certified robustness under Levenshtein distance.
Reviewers considered this a strong work — the problem is well-motivated and the technical methods overcome challenges from prior work. The proof involves bounding the Lipschitz constant of a network, and requires careful relaxations and estimates to handle the Levenshtein distance. The certificate is deterministic, in contrast to many other robustness certificates which are inherently randomized.
All reviewers recommend this for acceptance, and I agree.

**Additional Comments On Reviewer Discussion:**

See above.

---

### Decision · Program_Chairs · 2025-01-22

Accept (Poster)